# DEPTH PRO: SHARP MONOCULAR METRIC DEPTH IN LESS THAN A SECOND

**Aleksei Bochkovskii**      **Amaël Delaunoy**      **Hugo Germain**      **Marcel Santos**

**Yichao Zhou**      **Stephan R. Richter**      **Vladlen Koltun**

Apple

## ABSTRACT

We present a foundation model for zero-shot metric monocular depth estimation. Our model, Depth Pro, synthesizes high-resolution depth maps with unparalleled sharpness and high-frequency details. The predictions are metric, with absolute scale, without relying on the availability of metadata such as camera intrinsics. And the model is fast, producing a 2.25-megapixel depth map in 0.3 seconds on a standard GPU. These characteristics are enabled by a number of technical contributions, including an efficient multi-scale vision transformer for dense prediction, a training protocol that combines real and synthetic datasets to achieve high metric accuracy alongside fine boundary tracing, dedicated evaluation metrics for boundary accuracy in estimated depth maps, and state-of-the-art focal length estimation from a single image. Extensive experiments analyze specific design choices and demonstrate that Depth Pro outperforms prior work along multiple dimensions. We release code & weights at `https://github.com/apple/ml-depth-pro`

## 1 INTRODUCTION

Zero-shot monocular depth estimation underpins a growing variety of applications, such as advanced image editing, view synthesis, and conditional image generation. Inspired by MiDaS (Ranftl et al., 2022) and many follow-up works (Ranftl et al., 2021; Ke et al., 2024; Yang et al., 2024a; Piccinelli et al., 2024; Hu et al., 2024), applications increasingly leverage the ability to derive a dense pixelwise depth map for any image.

Our work is motivated in particular by novel view synthesis from a single image, an exciting application that has been transformed by advances in monocular depth estimation (Hedman et al., 2017; Shih et al., 2020; Jampani et al., 2021; Khan et al., 2023). Applications such as view synthesis imply a number of desiderata for monocular depth estimation. First, the depth estimator should work zero-shot on any image, not restricted to a specific domain (Ranftl et al., 2022; Yang et al., 2024a). Furthermore, the method should ideally produce *metric* depth maps in this zero-shot regime, to accurately reproduce object shapes, scene layouts, and absolute scales (Guizilini et al., 2023; Hu et al., 2024). For the broadest applicability 'in the wild', the method should produce metric depth maps with absolute scale even if no camera intrinsics (such as focal length) are provided with the image (Piccinelli et al., 2024). This enables view synthesis scenarios such as "Synthesize a view of this scene from 63 mm away" for essentially arbitrary single images (Dodgson, 2004).

Second, for the most compelling results, the monocular depth estimator should operate at high resolution and produce fine-grained depth maps that closely adhere to image details such as hair, fur, and other fine structures (Miangoleh et al., 2021; Ke et al., 2024; Li et al., 2024a). One benefit of producing sharp depth maps that accurately trace intricate details is the elimination of "flying pixels", which can degrade image quality in applications such as view synthesis (Jampani et al., 2021).

Third, for many interactive application scenarios, the depth estimator should operate at low latency, processing a high-resolution image in less than a second to support interactive view synthesis "queries" on demand. Low latency is a common characteristic of methods that reduce zero-shot monocular depth estimation to a single forward pass through a neural network (Ranftl et al., 2021; Yang et al., 2024a;

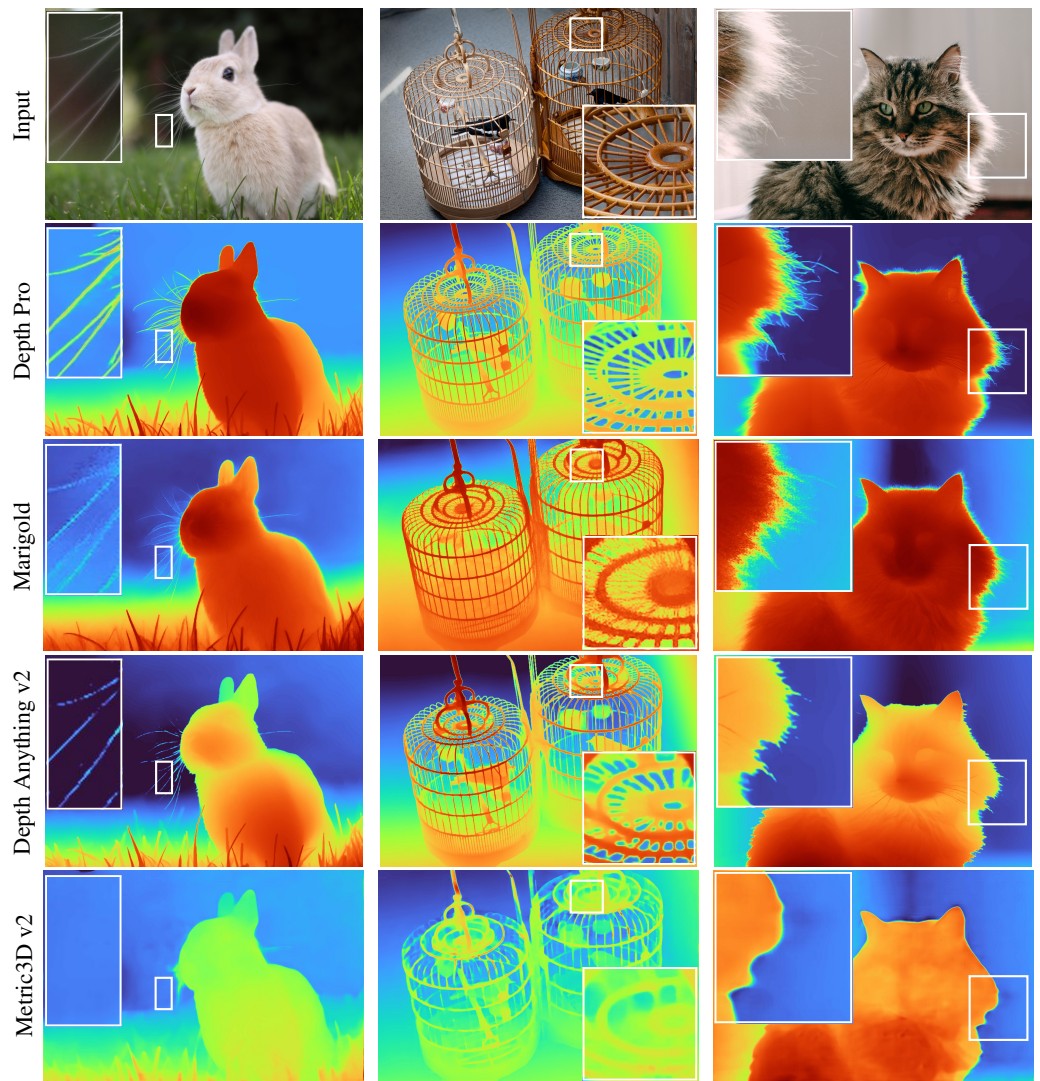

Figure 1: Results on images from the AM-2k (Li et al., 2022a) (1st & 3rd column) and DIS-5k (Qin et al., 2022) (2nd column) datasets. Input image on top, estimated depth maps from Depth Pro, Marigold (Ke et al., 2024), Depth Anything v2 (Yang et al., 2024b), and Metric3D v2 (Hu et al., 2024) below. Depth Pro produces zero-shot metric depth maps with absolute scale at 2.25-megapixel native resolution in 0.3 seconds on a V100 GPU.

Piccinelli et al., 2024), but it is not always shared by methods that employ more computationally demanding machinery at test time (Ke et al., 2024; Li et al., 2024a).

In this work, we present a foundation model for zero-shot metric monocular depth estimation that meets all of these desiderata. Our model, Depth Pro, produces metric depth maps with absolute scale on arbitrary images 'in the wild' without requiring metadata such as camera intrinsics. It operates at high resolution, producing 2.25-megapixel depth maps (with a native output resolution of $1536 \times 1536$ before optional further upsampling) in 0.3 seconds on a V100 GPU. Fig. 1 shows some representative results. Depth Pro dramatically outperforms all prior work in sharp delineation of object boundaries,

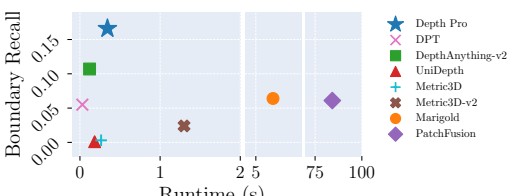

Figure 2: Boundary recall versus runtime. Depth Pro outperforms prior work by a multiplicative factor in boundary accuracy while being orders of magnitude faster than works focusing on fine-grained predictions (e.g., Marigold, PatchFusion).

including fine structures such as hair, fur, and vegetation. As shown in Fig. 2, Depth Pro offers unparalleled boundary tracing, outperforming all prior work by a multiplicative factor in boundary recall. (See Sec. 4 for additional detail.) Compared to the prior state of the art in boundary accuracy (Ke et al., 2024; Li et al., 2024a), Depth Pro is one to two orders of magnitude faster, yields much more accurate boundaries, and provides metric depth maps with absolute scale.

Depth Pro is enabled by a number of technical contributions. First, we derive a new set of metrics that enable leveraging highly accurate matting datasets for quantifying the accuracy of boundary tracing in evaluating monocular depth maps. We analyze the effect of typical output resolutions and find that a high resolution is necessary but not sufficient to improve boundary accuracy. Second, we design an efficient multi-scale ViT-based architecture for capturing the global image context while also adhering to fine structures at high resolution. Third, we devise a set of loss functions and a training curriculum that promote sharp depth estimates while training on real-world datasets that provide coarse and inaccurate supervision around boundaries, along with synthetic datasets that offer accurate pixelwise ground truth but limited realism. Fourth, we contribute zero-shot focal length estimation from a single image that dramatically outperforms the prior state of the art.

## 2    RELATED WORK

Early work on monocular depth estimation focused on training on individual datasets recorded with a single camera (Saxena et al., 2009; Eigen et al., 2014; Eigen & Fergus, 2015). Although this setup directly enabled metric depth predictions, it was limited to single datasets and narrow domains.

**Zero-shot depth estimation.** MegaDepth (Li & Snavely, 2018) demonstrated that training on a diverse dataset allows generalizing monocular depth prediction beyond a specific domain. Mi-DaS (Ranftl et al., 2022) advanced this idea by training on a large mix of diverse datasets with a scale-and-shift-invariant loss. Follow-up works applied this recipe to transformer-based architectures (Ranftl et al., 2021; Birkl et al., 2023) and further expanded the set of feasible datasets through self-supervision (Spencer et al., 2023; Yang et al., 2024a). A line of work uses self-supervision to learn from unlabeled image and video data (Petrovai & Nedevschi, 2022; Yang et al., 2024a). A number of recent approaches (Ke et al., 2024; Gui et al., 2025) harness diffusion models to synthesize relative depth maps. Although some of these methods demonstrated excellent generalization, their predictions are ambiguous in scale and shift, which precludes downstream applications that require accurate shapes, sizes, or distances.

**Zero-shot metric depth.** A line of work sought to improve metric depth prediction through a global distribution of depth values (Fu et al., 2018; Bhat et al., 2021; 2022; Li et al., 2024b) and further conditioning on scene type (Bhat et al., 2023). A different approach directly takes into account camera intrinsics. Cam-Convs (Fácil et al., 2019) conditioned convolutions on the camera intrinsics. LeReS (Yin et al., 2021) trains a separate network for undistorting point clouds to recover scale and shift, Metric3D (Yin et al., 2023) scales images or depth maps to a canonical space and remaps estimated depth given the focal length, and ZeroDepth (Guizilini et al., 2023) learns camera-specific embedddings in a variational framework. DMD (Saxena et al., 2023) conditions a diffusion model on the field of view. Metric3D v2 (Hu et al., 2024) leverages surface normals as an auxilliary output to improve metric depth. All of these methods require the camera intrinsics to be known and accurate. More recent works attempt to reason about unknown camera intrinsics either through a separate network (Spencer et al., 2024) or by predicting a camera embedding for conditioning its depth predictions in a spherical space (Piccinelli et al., 2024). Akin to these recent approaches, our method does not require the focal length to be provided as input. We propose to directly estimate the field of view from intermediate features of the depth prediction network, and show that this substantially outperforms the prior state of the art in the task of cross-domain focal length estimation.

**Sharp occluding contours.** SharpNet (Ramamonjisoa & Lepetit, 2019) incorporates normals and occluding contour constraints, but requires additional contour and normal supervision during training. BoostingDepth (Miangoleh et al., 2021) obtains detailed predictions from a low-resolution network by applying it independently to image patches. Since the patches lack global context, BoostingDepth fuses them through a sophisticated multi-step pipeline. PatchFusion (Li et al., 2024a) refines this concept through image-adaptive patch sampling and tailored modules that enable end-to-end training. A recent line of work (Gui et al., 2025; Ke et al., 2024) leverages diffusion priors to enhance the

sharpness of occlusion boundaries. These approaches predominantly focus on predicting relative (rather than metric) depth. We propose a simpler architecture without task-specific modules or diffusion priors and demonstrate that even sharper and more accurate results can be obtained while producing metric depth maps and reducing runtime by more than two orders of magnitude.

Guided depth super-resolution uses the input image to upsample low-resolution depth predictions (Metzger et al., 2023; Zhong et al., 2023). SMDNet (Tosi et al., 2021) predicts bimodal mixture densities to sharpen occluding contours. And Ramamonjisoa et al. (Ramamonjisoa et al., 2020) introduce a module for learning to sharpen depth boundaries of a pretrained network. These works are orthogonal to ours and could be applied to further upsample our high-resolution predictions.

To evaluate boundary tracing in predicted depth maps, Koch et al. (2018) introduce the iBims dataset with manual annotations of occluding contours and corresponding metrics. The need for manual annotation and highly accurate depth ground truth constrain the benchmark to a small set of indoor scenes. We contribute metrics based on segmentation and matting datasets that provide a complementary view by enabling evaluation on complex, dynamic environments or scenes with exceedingly fine detail for which ground-truth depth is impossible to obtain.

**Multi-scale vision transformers.** Vision transformers (ViTs) have emerged as the dominant general-purpose architecture for perception tasks but operate at low resolution (Dosovitskiy et al., 2021). The computational complexity of their attention layers prohibits naïvely scaling them to higher resolutions and several works proposed alternatives (Zhu et al., 2021; Liu et al., 2021; Li et al., 2022c; Chu et al., 2021; Liu et al., 2022a; 2023; Cai et al., 2023; Jaegle et al., 2022). Another line of work modified the ViT architecture to produce a hierarchy of features (Fan et al., 2021; Xie et al., 2021; Yuan et al., 2021; Ranftl et al., 2021; Chen et al., 2021; Lee et al., 2022).

Rather than modifying the ViT architecture, which requires computationally expensive retraining, we propose an architecture that applies a plain ViT backbone at multiple scales and fuses predictions into a single high-resolution output. This design benefits from ongoing improvements in ViT pretraining, as new variants can be easily swapped in (Oquab et al., 2024; Peng et al., 2022b; Sun et al., 2023).

Pretrained vision transformers have been adapted for semantic segmenation and object detection. ViT-Adapter (Chen et al., 2023) and ViT-CoMer (Xia et al., 2024) supplement a pretrained ViT with a convolutional network for dense prediction, whereas ViT-Det (Li et al., 2022b) builds a feature pyramid on top of a pretrained ViT. Distinct from these, we fuse features from the ViT applied at multiple scales to learn global context together with local detail.

## 3 METHOD

### 3.1 NETWORK

The key idea behind our network is to apply plain ViT encoders (Dosovitskiy et al., 2021) on patches extracted at multiple scales and fuse their predictions into a single high-resolution dense depth prediction in an end-to-end trainable model (see Fig. 3). As the patch encoder shares weights across all scales, it may intuitively learn a scale-invariant representation. The image encoder anchors the patch predictions in a global context. It is applied to the whole input image, downsampled to the base input resolution of the chosen encoder backbone (in our case $384 \times 384$).

The whole network operates at a fixed resolution of $1536 \times 1536$, which was chosen as a multiple of the ViT's $384 \times 384$. This guarantees a sufficiently large receptive field and constant runtimes for any image while preventing out-of-memory errors (which we repeatedly observed for variable-resolution approaches on large images). Confirming this design choice, our results in Sec. 4 and Tab. 5 demonstrate that Depth Pro is consistently orders of magnitude faster than variable-resolution approaches while being more accurate and producing sharper boundaries. A key benefit of assembling our architecture from plain ViT encoders over custom encoders is the abundance of pretrained ViT-based backbones that can be harnessed (Oquab et al., 2024; Peng et al., 2022b; Sun et al., 2023). We evaluate several pretrained backbones and compare our architecture to other high-resolution architectures in the appendices (Tab. 9 and Sec. B.3).

After initial downsampling to $1536 \times 1536$, the input image is split into patches of $384 \times 384$ at each scale. For the two finest scales, we let patches overlap to avoid seams, which yields 25 and 9

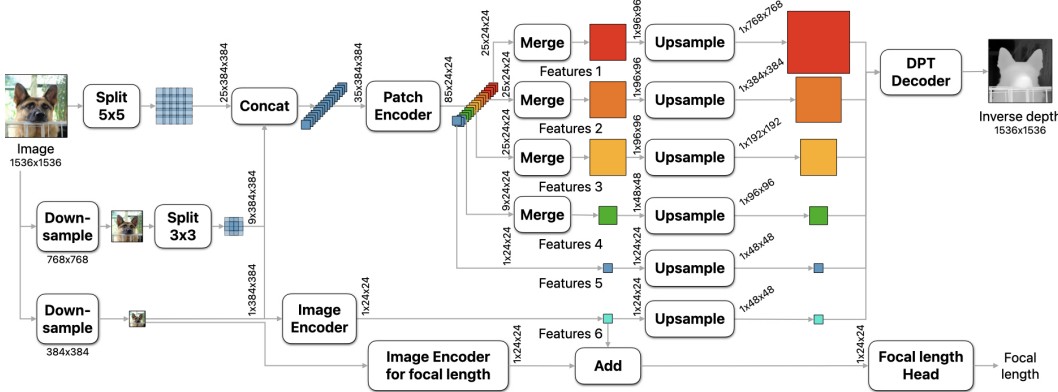

Figure 3: Overview of the network architecture. An image is downsampled at several scales. At each scale, it is split into patches, which are processed by a ViT-based patch encoder, with weights shared across scales. Patches are merged into feature maps, upsampled, and fused via a DPT decoder. Predictions are anchored by a separate image encoder that provides global context.

patches, respectively. In total, we extract 35 patches, concatenate them along the batch dimension to allow efficient batch processing, and feed them to the patch encoder. This yields a feature tensor at resolution $24 \times 24$ per input patch (Features 3 – 6 in Fig. 3). At the finest scale we further extract intermediate features (Features 1 & 2 in Fig. 3) to capture finer-grained details, yielding additional $25 + 25 = 50$ feature patches. We merge the feature patches into maps (detailed in Sec. C.1), which are fed into the decoder module, which resembles the DPT decoder (Ranftl et al., 2021).

In addition to sharing representations across scales, the patch-based application of the encoder network allows trivial parallelization as patches can be processed independently. Another source of computational efficiency comes from the lower computational complexity of patch-based processing in comparison to scaling up the ViT to higher resolutions. The reason is multi-head self-attention (Vaswani et al., 2017), whose computational complexity scales quadratically with the number of input pixels, and thus quartically in image dimension.

## 3.2 SHARP MONOCULAR DEPTH ESTIMATION

**Training objectives.** For each input image $I$, our network $f$ predicts a canonical inverse depth image $C = f(I)$. To obtain a dense metric depth map $D_m$, we scale by the horizontal field of view, represented by the focal length $f_{px}$ and the width $w$ (Yin et al., 2023): $D_m = \frac{f_{px}}{wC}$.

We train with several objectives, all based on canonical inverse depth, because this prioritizes areas close to the camera over farther areas or the whole scene, and thus supports visual quality in applications such as novel view synthesis (see Sec. B.4). Let $\hat{C}$ be the ground-truth canonical inverse depth. For all metric datasets we compute the mean absolute error ($\mathcal{L}_{MAE}$, Eq. 1) per pixel $i$, and discard pixels with an error in the top $20\%$ per image for real-world (as opposed to synthetic) datasets:

$$\mathcal{L}_{MAE}(\hat{C}, C) = \frac{1}{N} \sum_i^N |\hat{C}_i - C_i|. \tag{1}$$

For all non-metric datasets (i.e., those without reliable camera intrinsics or inconsistent scale), we normalize predictions and ground truth via the mean absolute deviation from the median (Ranftl et al., 2022) before applying a loss. We further compute errors on the first and second derivatives of (canoncial) inverse depth maps at multiple scales. Let $\nabla_*$ indicate a spatial derivative operator $*$, such as Scharr (S) (Scharr et al., 1997) or Laplace (L), and $p$ the error norm. We define the multi-scale derivative loss over $M$ scales as

$$\mathcal{L}_{*,p,M}(C, \hat{C}) = \frac{1}{M} \sum_j^M \frac{1}{N_j} \sum_i^{N_j} |\nabla_* C_i^j - \nabla_* \hat{C}_i^j|^p, \tag{2}$$

where the scales $j$ are computed by blurring and downsampling the inverse depth maps by a factor of 2 per scale. As shorthands we define the Mean Absolute Gradient Error $\mathcal{L}_{MAGE} = \mathcal{L}_{S,1,6}$, the Mean Absolute Laplace Error $\mathcal{L}_{MALE} = \mathcal{L}_{L,1,6}$, and the Mean Squared Gradient Error $\mathcal{L}_{MSGE} = \mathcal{L}_{S,2,6}$.

**Training curriculum.** We propose a training curriculum motivated by the following observations. First, training on a large mix of real-world and synthetic datasets improves generalization as measured by zero-shot accuracy (Ranftl et al., 2022; 2021; Yang et al., 2024a; Hu et al., 2024). Second, synthetic datasets provide pixel-accurate ground truth, whereas real-world datasets often contain missing areas, mismatched depth, or false measurements on object boundaries. Third, predictions get sharper over the course of training.

Based on these observations, we design a two-stage training curriculum. In the first stage, we aim to learn robust features that allow the network to generalize across domains. To that end, we train on a mix of all labeled training sets. Specifically, we minimize $\mathcal{L}_{MAE}$ on metric datasets and its normalized version on non-metric datasets. $\mathcal{L}_{MAE}$ is chosen for its robustness in handling potentially corrupted real-world ground truth. To steer the network towards sharp boundaries, we aim to also supervise on gradients of the predictions. Done naïvely, however, this can hinder optimization and slow down convergence. We found that a scale-and-shift-invariant loss on gradients, applied only to synthetic datasets, worked best. Controlled experiments are reported in Sec. B.5 of the appendices.

The second stage of training is designed to sharpen boundaries and reveal fine details in the predicted depth maps. To minimize the effect of inaccurate ground truth, at this stage we only train on synthetic datasets that provide high-quality pixel-accurate ground truth. (Note that this inverts the common practice of first training on synthetic data and then fine-tuning on real data (Gaidon et al., 2016; Gómez et al., 2023; Sun et al., 2021).) Specifically, we again minimize the $\mathcal{L}_{MAE}$ and supplement it with a selection of losses on the first- and second-order derivatives: $\mathcal{L}_{MAGE}$, $\mathcal{L}_{MALE}$, and $\mathcal{L}_{MSGE}$. We provide a detailed specification of the loss functions that are applied at each stage in the appendices.

**Evaluation metrics for sharp boundaries.** Applications such as novel view synthesis require depth maps to adhere to object boundaries. This is particularly challenging for thin structures. Misaligned or blurry boundaries can make objects appear distorted or split into parts. Common benchmarks for monocular depth prediction rarely take boundary sharpness into account. This may be attributed in part to the lack of diverse and realistic datasets with precise pixel-accurate ground-truth depth. To address this shortcoming, we propose a new set of metrics specifically for the evaluation of depth boundaries. Our key observation is that we can leverage existing high-quality annotations for matting, saliency, or segmentation as ground truth for depth boundaries. We treat annotations for these tasks as binary maps, which define a foreground/background relationship between an object and its environment. (This relationship may not hold in every case, especially for segmentation masks. However, we can easily discard such problematic cases through manual inspection. It is much easier to filter out a segmentation mask than to annotate it.) To ensure that the relationship holds, we only consider pixels around edges in the binary map.

We first define the metrics for depth maps and later derive the formulation for binary segmentation masks. Motivated by the ranking loss (Chen et al., 2016), we use the pairwise depth ratio of neighboring pixels to define a foreground/background relationship. Let $i, j$ be the locations of two neighboring pixels. We then define an occluding contour $c_d$ derived from a depth map $d$ as $c_d(i,j) = \left[ \frac{d(j)}{d(i)} > (1 + \frac{t}{100}) \right]$, where $[\cdot]$ is the Iverson bracket. Intuitively, this indicates the presence of an occluding contour between pixels $i$ and $j$ if their corresponding depth differs by more than $t\%$. For all pairs of neighboring pixels, we can then compute the precision ($P$) and recall ($R$) as

$$\text{P}(t) = \frac{\sum_{i,j \in N(i)} c_d(i,j) \wedge c_{\hat{d}}(i,j)}{\sum_{i,j \in N(i)} c_d(i,j)} \text{ and } \text{R}(t) = \frac{\sum_{i,j \in N(i)} c_d(i,j) \wedge c_{\hat{d}}(i,j)}{\sum_{i,j \in N(i)} c_{\hat{d}}(i,j)}. \quad (3)$$

Note that both $P$ and $R$ are scale-invariant. In our experiments, we report the F1 score. To account for multiple relative depth ratios, we further perform a weighted averaging of the F1 values with thresholds that range linearly from $t_{min} = 5$ to $t_{max} = 25$, with stronger weights towards high threshold values. Compared to other edge-based metrics (such as the edge accuracy and completion from iBims (Koch et al., 2018)), our metric does not require any manual edge annotation, but simply pixelwise ground truth, which is easily obtained for synthetic datasets.

Similarly, we can also identify occluding contours from binary label maps that can be derived from real-world segmentation, saliency, and matting datasets. Given a binary mask $b$ over the image, we define the presence of an occluding contour $c_b$ between pixels $i, j$ as $c_b(i, j) = b(i) \wedge \neg b(j)$. With this definition at hand, we compute the recall $R(t)$ by replacing the occluding contours from depth maps in Eq. 3 with those from binary maps. Since the binary maps commonly label whole objects, we cannot obtain ground-truth occluding contours that do not align with object silhouettes. Thus the boundary annotation is incomplete – some but not all occluding contours are identified by this procedure. Therefore we can only compute the recall but not the precision for binary maps.

To penalize blurry edges, we suppress non-maximum values of $c_{\hat{d}}$ within the valid bounds of $c_{\hat{d}}(i, j)$ connected components. We report additional experiments and qualitative results in Sec. A.4.

### 3.3 FOCAL LENGTH ESTIMATION

To handle images that may have inaccurate or missing `EXIF` metadata, we supplement our network with a focal length estimation head. A small convolutional head ingests frozen features from the depth estimation network and task-specific features from a separate ViT image encoder to predict the horizontal angular field-of-view. We use $\mathcal{L}_2$ as the training loss. We train the focal length head and the ViT encoder after the depth estimation training. Separating the focal length training has several benefits over joint training with the depth network. It avoids the necessity of balancing the depth and focal length training objectives. It also allows training the focal length head on a different set of datasets, excluding some narrow-domain single-camera datasets that are used in training the depth estimation network, and adding large-scale image datasets that provide focal length supervision but no depth supervision. Further details are provided in Sec. C.2.

## 4 EXPERIMENTS

This section summarizes the key results. Additional details and experiments are reported in the appendices, including details on datasets, hyperparameters, experimental protocols, and the comparison of runtimes, which is summarized in Fig. 2. The appendices also report controlled experiments, including controlled studies on network architectures, training objectives, and training curricula.

Here we summarize a number of key comparisons of Depth Pro to state-of-the-art metric monocular depth estimation systems. One challenge in conducting such a comparison is that many leading recent systems are trained on bespoke combinations of datasets. Some systems use proprietary datasets that are not publicly available, and some use datasets that are only available under restrictive licenses. Some recent systems also train on unlabeled datasets or incorporate pretrained models (e.g., diffusion models) that were trained on additional massive datasets. This rules out the possibility of a comparison that controls for training data (e.g., only comparing to systems that use the same datasets we do). At this stage of this research area, the only feasible comparison to other leading cross-domain monocular depth estimation models is on a full system-to-system basis. Fully trained models (each trained on a large, partially overlapping and partially distinct collection of datasets) are compared to each other zero-shot on datasets that none of the compared systems trained on.

**Zero-shot metric depth.** We evaluate our method's ability to predict zero-shot *metric* depth and compare against the state of the art in Tab. 1. Our baselines include Depth Anything (Yang et al., 2024a), Metric3D (Yin et al., 2023), PatchFusion (Li et al., 2024a), UniDepth (Piccinelli et al., 2024), ZeroDepth (Guizilini et al., 2023) and ZoeDepth (Bhat et al., 2023). We also report results for the very recent Depth Anything v2 (Yang et al., 2024b) and Metric3D v2 (Hu et al., 2024).

As an overall summary measure of metric depth accuracy, Tab. 1 uses the $\delta_1$ metric (Ladicky et al., 2014), which is commonly used for this purpose (Yin et al., 2023; Yang et al., 2024a; Piccinelli et al., 2024). It is defined as the percentage of inlier pixels, for which the predicted and ground-truth depths are within 25% of each other. We picked this metric for its robustness, with the strictest threshold found in the literature (25%).

Corresponding tables for additional metrics can be found in Sec. A.2 of the appendices, including *AbsRel* (Ladicky et al., 2014), $Log_{10}$, $\delta_2$ and $\delta_3$ scores, as well as point-cloud metrics (Spencer et al., 2022). Tab. 1 also reports the average rank of each method across datasets, a common way to summarize cross-dataset performance (Ranftl et al., 2022).

Table 1: **Zero-shot metric depth accuracy.** We report the $\delta_1$ score per dataset (higher is better) and aggregate performance across datasets via the average rank (lower is better). Methods in gray are not strictly zero-shot. Results on additional metrics and datasets are presented in the appendices.

| Method | Booster | ETH3D | Middlebury | NuScenes | Sintel | Sun-RGBD | Avg. Rank↓ |
|---|---|---|---|---|---|---|---|
| DepthAnything (Yang et al., 2024a) | 52.3 | 9.3 | 39.3 | 35.4 | 6.9 | 85.0 | 5.7 |
| DepthAnything v2 (Yang et al., 2024b) | 59.5 | 36.3 | 37.2 | 17.7 | 5.9 | 72.4 | 5.8 |
| Metric3D (Yin et al., 2023) | 4.7 | 34.2 | 13.6 | 64.4 | 17.3 | 16.9 | 5.8 |
| Metric3D v2 (Hu et al., 2024) | 39.4 | 87.7 | 29.9 | 82.6 | 38.3 | 75.6 | 3.7 |
| PatchFusion (Li et al., 2024a) | 22.6 | 51.8 | 49.9 | 20.4 | 14.0 | 53.6 | 5.2 |
| UniDepth (Piccinelli et al., 2024) | 27.6 | 25.3 | 31.9 | **83.6** | 16.5 | **95.8** | 4.2 |
| ZeroDepth (Guizilini et al., 2023) | OOM | OOM | 46.5 | 64.3 | 12.9 | OOM | 4.6 |
| ZoeDepth (Bhat et al., 2023) | 21.6 | 34.2 | 53.8 | 28.1 | 7.8 | 85.7 | 5.3 |
| Depth Pro (Ours) | 46.6 | 41.5 | **60.5** | 49.1 | **40.0** | 89.0 | **2.5** |

Table 2: **Zero-shot boundary accuracy.** We report the F1 score for dataset with ground-truth depth, and boundary recall ($R$) for matting and segmentation datasets. Qualitative results are shown on a sample from the AM-2k dataset (Li et al., 2022a). Higher is better for all metrics.

| | Method | Sintel F1↑ | Spring F1↑ | iBims F1↑ | AM R↑ | P3M R↑ | DIS R↑ |
|---|---|---|---|---|---|---|---|
| Absolute | DPT (Ranftl et al., 2021) | 0.181 | 0.029 | 0.113 | 0.055 | 0.075 | 0.018 |
| | Metric3D (Yin et al., 2023) | 0.037 | 0.000 | 0.055 | 0.003 | 0.003 | 0.001 |
| | Metric3D v2 (Hu et al., 2024) | 0.321 | 0.024 | 0.096 | 0.024 | 0.013 | 0.006 |
| | ZoeDepth (Bhat et al., 2023) | 0.027 | 0.001 | 0.035 | 0.008 | 0.004 | 0.002 |
| | PatchFusion (Li et al., 2024a) | 0.312 | 0.032 | 0.134 | 0.061 | 0.109 | 0.068 |
| | UniDepth (Piccinelli et al., 2024) | 0.316 | 0.000 | 0.039 | 0.001 | 0.003 | 0.000 |
| Rel. | DepthAnything (Yang et al., 2024a) | 0.261 | 0.045 | 0.127 | 0.058 | 0.094 | 0.023 |
| | DepthAnything v2 (Yang et al., 2024b) | 0.228 | 0.056 | 0.111 | 0.107 | 0.131 | 0.056 |
| | Marigold (Ke et al., 2024) | 0.068 | 0.032 | 0.149 | 0.064 | 0.101 | 0.049 |
| | Depth Pro (Ours) | **0.409** | **0.079** | **0.176** | **0.173** | **0.168** | **0.077** |

Image     Alpha Matte     Depth Pro (Ours)     DepthAnything v2     PatchFusion     Marigold

We report results on Booster (Ramirez et al., 2024), Middlebury (Scharstein et al., 2014), Sun-RGBD (Song et al., 2015), ETH3D (Schöps et al., 2017), nuScenes (Caesar et al., 2020), and Sintel (Butler et al., 2012), because, to our knowledge, they were never used in training any of the evaluated systems. Despite our best efforts, we were not able to run ZeroDepth on Booster, Middlebury, or Sun-RGBD as it consistently ran out of memory due to the high image resolutions. More details on our evaluation setup can be found in Sec. C of the appendix.

The results in Tab. 1 confirm the findings of Piccinelli et al. (2024), who observed considerable domain bias in some of the leading metric depth estimation models. Notably, Depth Anything v1 & v2 focus on *relative* depth estimation; for metric depth, they provide different models for different domains, fine-tuned either for indoor or for outdoor scenes. Metric3D v1 & v2 provide domain-invariant models, but their performance depends strongly on careful selection of the crop size at test time, which is performed *per domain* in their experiments and thus violates the zero-shot premise. We tried setting the crop size automatically based on the aspect ratio of the image, but this substantially degraded the performance of Metric3D; for this reason, we use the recommended non-zero-shot protocol, with the recommended per-domain crop sizes. Since domain-specific models and crop sizes violate the strict zero-shot premise we (and other baselines) operate under, we mark the Depth Anything and Metric3D results in gray in Tab. 1.

We find that Depth Pro demonstrates the strongest generalization by consistently scoring among the top approaches per dataset and obtaining the best average rank across all datasets.

**Zero-shot boundaries.** Tab. 2 summarizes the evaluation of boundary accuracy for Depth Pro and several baselines. This evaluation is conducted in a zero-shot setting: models are only evaluated on datasets that were not seen during training. Since our boundary metrics are scale-invariant, our baselines here also include methods that only predict relative (rather than absolute metric) depth. Our absolute baselines include Metric3D (Yin et al., 2023), Metric3D v2 ('giant' model) (Hu et al., 2024), PatchFusion (Li et al., 2024a), UniDepth (Piccinelli et al., 2024), and ZoeDepth (Bhat et al., 2023). We also report results for the relative variants of Depth Anything v1 & v2 (Yang et al., 2024a;b) because they yield sharper boundaries than their metric counterparts. Lastly, we include Marigold (Ke et al., 2024), a recent diffusion-based relative depth model that became popular due to its high-fidelity predictions. We use the boundary metrics introduced in Sec. 3.2, and report the average boundary F1 score for datasets with ground-truth depth, and boundary recall ($R$) for datasets with matting or segmentation annotations. For image matting datasets, a pixel is marked as occluding when the value of the alpha matte is above $0.1$.

The datasets include Sintel (Butler et al., 2012) and Spring (Mehl et al., 2023), which are synthetic. We also include the iBims dataset (Koch et al., 2018) which is often used specifically to evaluate depth boundaries, despite having low resolution. We refer to the appendices for a full slate of iBims-specific metrics. To evaluate high-frequency structures encountered in natural images (such as hair or fur), we use AM-2k (Li et al., 2022a) and P3M-10k (Li et al., 2021), which are high-resolution image matting datasets that were used to evaluate image matting models (Li et al., 2023). Additionally, we further report results on the DIS-5k (Qin et al., 2022) image segmentation dataset. This is an object segmentation dataset that provides highly accurate binary masks across diverse images. We manually remove samples in which the segmented object is occluded by foreground objects. Fig. 2 visually summarizes the boundary recall metric on the AM-2k dataset, as a function of runtime.

We find that Depth Pro produces more accurate boundaries than all baselines on all datasets, by a significant margin. As can be observed in Fig. 1, in the images in Tab. 2, and the additional results in Sec. A, the competitive metric accuracy of Metric3D v2 and Depth Anything v2 does not imply sharp boundaries. Depth Pro has a consistently higher recall for thin structures like hair and fur and yields sharper boundaries. This is also true in comparison to the diffusion-based Marigold, as well as PatchFusion, which operates at variable resolution. Note that Depth Pro is orders of magnitude faster than Marigold and PatchFusion (see Fig. 2 & Tab. 5). Fig. 4 demonstrates the benefits of sharp boundary prediction for novel view synthesis from a single image.

**Focal length estimation.** Prior work (Piccinelli et al., 2024; Kocabas et al., 2021; Baradad & Torralba, 2020) does not provide comprehensive systematic evaluations of focal length estimators on in-the-wild images. To address this, we curated a *zero-shot* test dataset. To this end, we selected diverse datasets with intact EXIF data, enabling reliable assessment of focal length estimation accuracy. FiveK (Bychkovsky et al., 2011), DDDP (Abuolaim & Brown, 2020), and RAISE (Dang-Nguyen et al., 2015) contribute professional-grade photographs taken with SLR cameras. SPAQ (Fang et al., 2020) provides casual photographs from mobile phones. PPR10K (Liang et al., 2021) provides high-quality portrait images. Finally, ZOOM (Zhang et al., 2019) includes sets of scenes captured at various optical zoom levels.

Table 3: **Comparison on focal length estimation.** We report $\delta_{25\%}$ and $\delta_{50\%}$ for each dataset, i.e., the percentage of images with relative error (focal length in mm) less than 25% and 50%, respectively.

| | DDDP | | FiveK | | PPR10K | | RAISE | | SPAQ | | ZOOM | |
|---|---|---|---|---|---|---|---|---|---|---|---|---|
| | $\delta_{25\%}$ | $\delta_{50\%}$ | $\delta_{25\%}$ | $\delta_{50\%}$ | $\delta_{25\%}$ | $\delta_{50\%}$ | $\delta_{25\%}$ | $\delta_{50\%}$ | $\delta_{25\%}$ | $\delta_{50\%}$ | $\delta_{25\%}$ | $\delta_{50\%}$ |
| UniDepth (Piccinelli et al., 2024) | 6.8 | 40.3 | 24.8 | 56.2 | 13.8 | 44.2 | 35.4 | 74.8 | 44.2 | 77.4 | 20.4 | 45.4 |
| SPEC (Kocabas et al., 2021) | 14.6 | 46.3 | 30.2 | 56.6 | 34.6 | 67.0 | 49.2 | 78.6 | 50.0 | 82.2 | 23.2 | 43.6 |
| im2pcl (Baradad & Torralba, 2020) | 7.3 | 29.6 | 28.0 | 60.0 | 24.2 | 61.4 | 51.8 | 75.2 | 26.6 | 55.0 | 22.4 | 42.8 |
| Depth Pro (Ours) | **66.9** | **85.8** | **74.2** | **92.4** | **64.6** | **88.8** | **84.2** | **96.4** | **68.4** | **85.2** | **69.8** | **91.6** |

Tab. 3 compares Depth Pro against state-of-the-art focal length estimators and shows the percentage of images with relative estimation error under 25% and 50%, respectively. Depth Pro is the most accurate across all datasets. For example, on PPR10K, a dataset of human portraits, our method leads with 64.6% of the images having a focal length error below 25%, while the second-best method, SPEC, only achieves 34.6% on this metric. We attribute this superior performance to our network design and training protocol, which decouple training of the focal length estimator from the depth

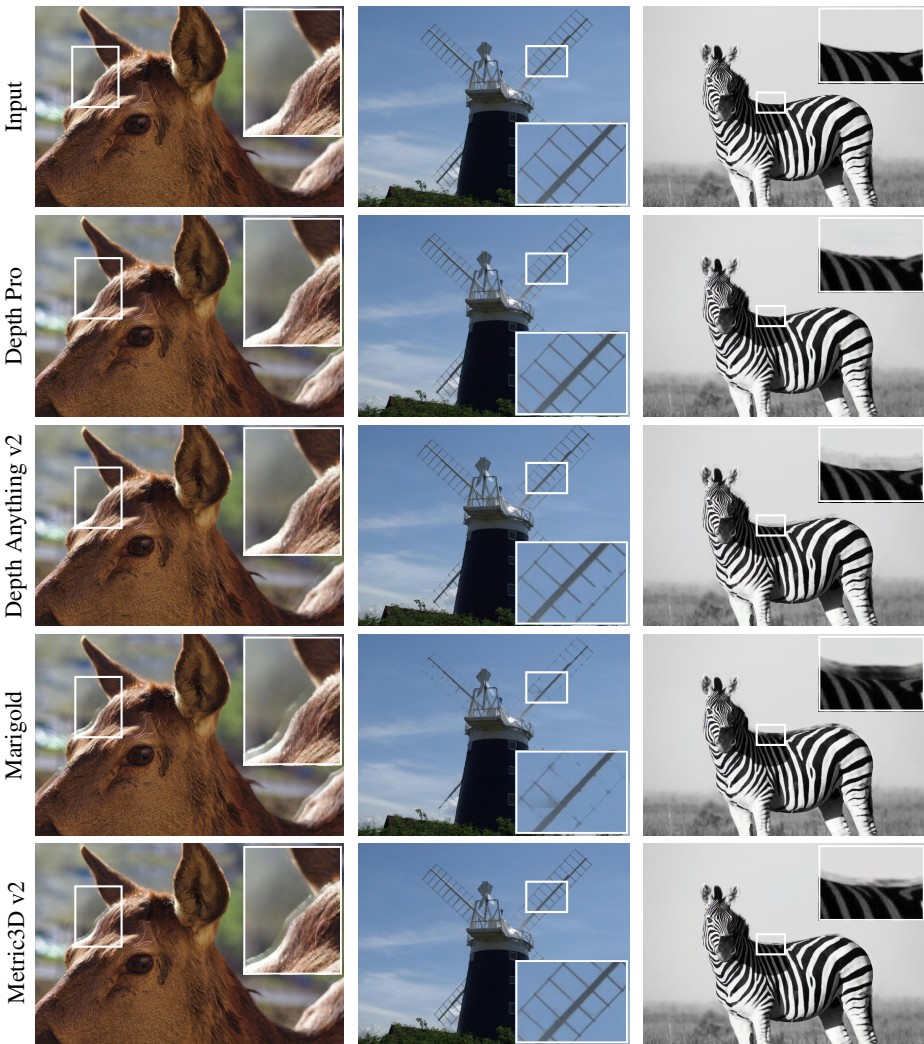

Figure 4: **Impact on novel view synthesis.** We plug depth maps produced by Depth Pro, Marigold (Ke et al., 2024), Depth Anything v2 (Yang et al., 2024b), and Metric3D v2 (Hu et al., 2024) into a recent publicly available novel view synthesis system (Khan et al., 2023). We demonstrate results on images from AM-2k (Li et al., 2022a) (1st & 3rd column) and DIS-5k (Qin et al., 2022) (2nd column). The insets highlight typical artifacts from inaccurate boundaries. All methods except Depth Pro add ghosting edges to the horse (1st column) and zebra (3rd column). Marigold and Depth Anything v2 miss thin structures of the windmill (2nd column). Depth Pro produces sharper and more accurate depth maps, yielding cleaner synthesized views.

network, enabling us to use different training sets for these two tasks. Further controlled experiments are reported in Sec. B.7 of the appendices.

## 5 CONCLUSION & LIMITATIONS

Depth Pro produces high-resolution metric depth maps with high-frequency detail at sub-second runtimes. Our model achieves state-of-the-art zero-shot metric depth estimation accuracy without requiring metadata such as camera intrinsics, and traces out occlusion boundaries in unprecedented detail, facilitating applications such as novel view synthesis from single images 'in the wild'. While Depth Pro outperforms prior work along multiple dimensions, it is not without limitations. For example, the model is limited in dealing with translucent surfaces and volumetric scattering, where the definition of pixelwise depth is ill-posed and ambiguous.

R E F E R E N C E S

Abdullah Abuolaim and Michael S Brown. Defocus deblurring using dual-pixel data. In *ECCV*, 2020.

Manel Baradad and Antonio Torralba. Height and uprightness invariance for 3D prediction from a single view. In *CVPR*, 2020.

Zuria Bauer, Francisco Gomez-Donoso, Edmanuel Cruz, Sergio Orts-Escolano, and Miguel Cazorla. UASOL, a large-scale high-resolution outdoor stereo dataset. *Scientific Data*, 6, 2019.

Shariq Farooq Bhat, Ibraheem Alhashim, and Peter Wonka. AdaBins: Depth estimation using adaptive bins. In *CVPR*, 2021.

Shariq Farooq Bhat, Ibraheem Alhashim, and Peter Wonka. LocalBins: Improving depth estimation by learning local distributions. In *ECCV*, 2022.

Shariq Farooq Bhat, Reiner Birkl, Diana Wofk, Peter Wonka, and Matthias Müller. ZoeDepth: Zero-shot transfer by combining relative and metric depth. *arXiv*, 2023.

Reiner Birkl, Diana Wofk, and Matthias Müller. MiDaS v3.1 - A model zoo for robust monocular relative depth estimation. *arXiv*, 2023.

Michael J. Black, Priyanka Patel, Joachim Tesch, and Jinlong Yang. BEDLAM: A synthetic dataset of bodies exhibiting detailed lifelike animated motion. In *CVPR*, 2023.

Daniel J. Butler, Jonas Wulff, Garrett B. Stanley, and Michael J. Black. A naturalistic open source movie for optical flow evaluation. In *ECCV*, 2012.

Vladimir Bychkovsky, Sylvain Paris, Eric Chan, and Frédo Durand. Learning photographic global tonal adjustment with a database of input / output image pairs. In *CVPR*, 2011.

Holger Caesar, Varun Bankiti, Alex H. Lang, Sourabh Vora, Venice Erin Liong, Qiang Xu, Anush Krishnan, Yu Pan, Giancarlo Baldan, and Oscar Beijbom. nuScenes: A multimodal dataset for autonomous driving. In *CVPR*, 2020.

Han Cai, Junyan Li, Muyan Hu, Chuang Gan, and Song Han. EfficientViT: Lightweight multi-scale attention for high-resolution dense prediction. In *ICCV*, 2023.

Chun-Fu (Richard) Chen, Quanfu Fan, and Rameswar Panda. CrossViT: Cross-attention multi-scale vision transformer for image classification. In *ICCV*, 2021.

Weifeng Chen, Zhao Fu, Dawei Yang, and Jia Deng. Single-image depth perception in the wild. In *NIPS*, 2016.

Zhe Chen, Yuchen Duan, Wenhai Wang, Junjun He, Tong Lu, Jifeng Dai, and Yu Qiao. Vision transformer adapter for dense predictions. In *ICLR*, 2023.

Xiangxiang Chu, Zhi Tian, Yuqing Wang, Bo Zhang, Haibing Ren, Xiaolin Wei, Huaxia Xia, and Chunhua Shen. Twins: Revisiting the design of spatial attention in vision transformers. In *NeurIPS*, 2021.

Angela Dai, Angel X. Chang, Manolis Savva, Maciej Halber, Thomas A. Funkhouser, and Matthias Nießner. ScanNet: Richly-annotated 3D reconstructions of indoor scenes. In *CVPR*, 2017.

Duc-Tien Dang-Nguyen, Cecilia Pasquini, Valentina Conotter, and Giulia Boato. RAISE: A raw images dataset for digital image forensics. In *MMSys*, 2015.

Afshin Dehghan, Gilad Baruch, Zhuoyuan Chen, Yuri Feigin, Peter Fu, Thomas Gebauer, Daniel Kurz, Tal Dimry, Brandon Joffe, Arik Schwartz, and Elad Shulman. ARKitScenes: A diverse real-world dataset for 3D indoor scene understanding using mobile RGB-D data. In *NeurIPS Datasets & Benchmarks*, 2021.

Jia Deng, Wei Dong, Richard Socher, Li-Jia Li, Kai Li, and Li Fei-Fei. ImageNet: A large-scale hierarchical image database. In *CVPR*, 2009.

Neil A. Dodgson. Variation and extrema of human interpupillary distance. In *Stereoscopic Displays and Virtual Reality Systems XI*, volume 5291, 2004.

Alexey Dosovitskiy, Lucas Beyer, Alexander Kolesnikov, Dirk Weissenborn, Xiaohua Zhai, Thomas Unterthiner, Mostafa Dehghani, Matthias Minderer, Georg Heigold, Sylvain Gelly, Jakob Uszkoreit, and Neil Houlsby. An image is worth 16x16 words: Transformers for image recognition at scale. In *ICLR*, 2021.

David Eigen and Rob Fergus. Predicting depth, surface normals and semantic labels with a common multi-scale convolutional architecture. In *ICCV*, 2015.

David Eigen, Christian Puhrsch, and Rob Fergus. Depth map prediction from a single image using a multi-scale deep network. In *NIPS*, 2014.

Facebook Research. fvcore. `https://github.com/facebookresearch/fvcore`, 2022.

José M. Fácil, Benjamin Ummenhofer, Huizhong Zhou, Luis Montesano, Thomas Brox, and Javier Civera. CAM-Convs: Camera-aware multi-scale convolutions for single-view depth. In *CVPR*, 2019.

Haoqi Fan, Bo Xiong, Karttikeya Mangalam, Yanghao Li, Zhicheng Yan, Jitendra Malik, and Christoph Feichtenhofer. Multiscale vision transformers. In *ICCV*, 2021.

Yuming Fang, Hanwei Zhu, Yan Zeng, Kede Ma, and Zhou Wang. Perceptual quality assessment of smartphone photography. In *CVPR*, 2020.

Huan Fu, Mingming Gong, Chaohui Wang, Kayhan Batmanghelich, and Dacheng Tao. Deep ordinal regression network for monocular depth estimation. In *CVPR*, 2018.

Adrien Gaidon, Qiao Wang, Yohann Cabon, and Eleonora Vig. Virtual worlds as proxy for multi-object tracking analysis. In *CVPR*, 2016.

Andreas Geiger, Philip Lenz, Christoph Stiller, and Raquel Urtasun. Vision meets robotics: The KITTI dataset. *IJRR*, 32(11), 2013.

Jose Luis Gómez, Manuel Silva, Antonio Seoane, Agnès Borràs, Mario Noriega, Germán Ros, José Antonio Iglesias Guitián, and Antonio M. López. All for one, and one for all: UrbanSyn dataset, the third musketeer of synthetic driving scenes. *arXiv*, 2023.

Ming Gui, Johannes S. Fischer, Ulrich Prestel, Pingchuan Ma, Dmytro Kotovenko, Olga Grebenkova, Stefan Andreas Baumann, Vincent Tao Hu, and Björn Ommer. DepthFM: Fast monocular depth estimation with flow matching. In *AAAI*, 2025.

Vitor Guizilini, Rares Ambrus, Sudeep Pillai, Allan Raventos, and Adrien Gaidon. 3D packing for self-supervised monocular depth estimation. In *CVPR*, 2020.

Vitor Guizilini, Igor Vasiljevic, Dian Chen, Rares Ambrus, and Adrien Gaidon. Towards zero-shot scale-aware monocular depth estimation. In *ICCV*, 2023.

Kaiming He, Xinlei Chen, Saining Xie, Yanghao Li, Piotr Dollár, and Ross Girshick. Masked autoencoders are scalable vision learners. In *CVPR*, 2022.

Peter Hedman, Suhib Alsisan, Richard Szeliski, and Johannes Kopf. Casual 3D photography. *ACM Trans. Graph.*, 36(6), 2017.

Mu Hu, Wei Yin, Chi Zhang, Zhipeng Cai, Xiaoxiao Long, Hao Chen, Kaixuan Wang, Gang Yu, Chunhua Shen, and Shaojie Shen. Metric3D v2: A versatile monocular geometric foundation model for zero-shot metric depth and surface normal estimation. *TPAMI*, 46(12), 2024.

Yuan-Ting Hu, Jiahong Wang, Raymond A. Yeh, and Alexander G. Schwing. SAIL-VOS 3D: A synthetic dataset and baselines for object detection and 3D mesh reconstruction from video data. In *CVPR*, 2021.

Po-Han Huang, Kevin Matzen, Johannes Kopf, Narendra Ahuja, and Jia-Bin Huang. DeepMVS: Learning multi-view stereopsis. In *CVPR*, 2018.

Xinyu Huang, Peng Wang, Xinjing Cheng, Dingfu Zhou, Qichuan Geng, and Ruigang Yang. The ApolloScape open dataset for autonomous driving and its application. *TPAMI*, 42(10), 2020.

Andrew Jaegle, Sebastian Borgeaud, Jean-Baptiste Alayrac, Carl Doersch, Catalin Ionescu, David Ding, Skanda Koppula, Daniel Zoran, Andrew Brock, Evan Shelhamer, Olivier J. Hénaff, Matthew M. Botvinick, Andrew Zisserman, Oriol Vinyals, and João Carreira. Perceiver IO: A general architecture for structured inputs & outputs. In *ICLR*, 2022.

Varun Jampani, Huiwen Chang, Kyle Sargent, Abhishek Kar, Richard Tucker, Michael Krainin, Dominik Kaeser, William T. Freeman, David Salesin, Brian Curless, and Ce Liu. SLIDE: Single image 3D photography with soft layering and depth-aware inpainting. In *ICCV*, 2021.

Nikita Karaev, Ignacio Rocco, Benjamin Graham, Natalia Neverova, Andrea Vedaldi, and Christian Rupprecht. DynamicStereo: Consistent dynamic depth from stereo videos. In *CVPR*, 2023.

Bingxin Ke, Anton Obukhov, Shengyu Huang, Nando Metzger, Rodrigo Caye Daudt, and Konrad Schindler. Repurposing diffusion-based image generators for monocular depth estimation. In *CVPR*, 2024.

Numair Khan, Lei Xiao, and Douglas Lanman. Tiled multiplane images for practical 3D photography. In *ICCV*, 2023.

Youngjung Kim, Bumsub Ham, Changjae Oh, and Kwanghoon Sohn. Structure selective depth superresolution for RGB-D cameras. *TIP*, 25(11), 2016.

Alexander Kirillov, Eric Mintun, Nikhila Ravi, Hanzi Mao, Chloe Rolland, Laura Gustafson, Tete Xiao, Spencer Whitehead, Alexander C. Berg, Wan-Yen Lo, Piotr Dollar, and Ross Girshick. Segment anything. In *ICCV*, 2023.

Muhammed Kocabas, Chun-Hao P. Huang, Joachim Tesch, Lea Müller, Otmar Hilliges, and Michael J. Black. SPEC: Seeing people in the wild with an estimated camera. In *ICCV*, 2021.

Tobias Koch, Lukas Liebel, Friedrich Fraundorfer, and Marco Körner. Evaluation of CNN-based single-image depth estimation methods. In *ECCV Workshops*, 2018.

Anastasiia Kornilova, Marsel Faizullin, Konstantin Pakulev, Andrey Sadkov, Denis Kukushkin, Azat Akhmetyanov, Timur Akhtyamov, Hekmat Taherinejad, and Gonzalo Ferrer. SmartPortraits: Depth powered handheld smartphone dataset of human portraits for state estimation, reconstruction and synthesis. In *CVPR*, 2022.

Lubor Ladicky, Jianbo Shi, and Marc Pollefeys. Pulling things out of perspective. In *CVPR*, 2014.

Hoang-An Le, Thomas Mensink, Partha Das, Sezer Karaoglu, and Theo Gevers. EDEN: Multimodal synthetic dataset of enclosed garden scenes. In *WACV*, 2021.

Youngwan Lee, Jonghee Kim, Jeffrey Willette, and Sung Ju Hwang. MPViT: Multi-path vision transformer for dense prediction. In *CVPR*, 2022.

Jizhizi Li, Sihan Ma, Jing Zhang, and Dacheng Tao. Privacy-preserving portrait matting. In *ACM-MM*, 2021.

Jizhizi Li, Jing Zhang, Stephen J Maybank, and Dacheng Tao. Bridging composite and real: Towards end-to-end deep image matting. *IJCV*, 130(2), 2022a.

Jizhizi Li, Jing Zhang, and Dacheng Tao. Deep image matting: A comprehensive survey. *arXiv*, 2023.

Yanghao Li, Hanzi Mao, Ross Girshick, and Kaiming He. Exploring plain vision transformer backbones for object detection. In *ECCV*, 2022b.

Yanghao Li, Chao-Yuan Wu, Haoqi Fan, Karttikeya Mangalam, Bo Xiong, Jitendra Malik, and Christoph Feichtenhofer. MViTv2: Improved multiscale vision transformers for classification and detection. In *CVPR*, 2022c.

Zhengqi Li and Noah Snavely. MegaDepth: Learning single-view depth prediction from internet photos. In *CVPR*, 2018.

Zhenyu Li, Shariq Farooq Bhat, and Peter Wonka. PatchFusion: An end-to-end tile-based framework for high-resolution monocular metric depth estimation. In *CVPR*, 2024a.

Zhenyu Li, Xuyang Wang, Xianming Liu, and Junjun Jiang. BinsFormer: Revisiting adaptive bins for monocular depth estimation. *TIP*, 33, 2024b.

Jie Liang, Hui Zeng, Miaomiao Cui, Xuansong Xie, and Lei Zhang. PPR10K: A large-scale portrait photo retouching dataset with human-region mask and group-level consistency. In *CVPR*, 2021.

Xinyu Liu, Houwen Peng, Ningxin Zheng, Yuqing Yang, Han Hu, and Yixuan Yuan. EfficientViT: Memory efficient vision transformer with cascaded group attention. In *CVPR*, 2023.

Ze Liu, Yutong Lin, Yue Cao, Han Hu, Yixuan Wei, Zheng Zhang, Stephen Lin, and Baining Guo. Swin transformer: Hierarchical vision transformer using shifted windows. In *ICCV*, 2021.

Ze Liu, Han Hu, Yutong Lin, Zhuliang Yao, Zhenda Xie, Yixuan Wei, Jia Ning, Yue Cao, Zheng Zhang, Li Dong, Furu Wei, and Baining Guo. Swin transformer V2: Scaling up capacity and resolution. In *CVPR*, 2022a.

Zhuang Liu, Hanzi Mao, Chao-Yuan Wu, Christoph Feichtenhofer, Trevor Darrell, and Saining Xie. A ConvNet for the 2020s. In *CVPR*, 2022b.

Lukas Mehl, Jenny Schmalfuss, Azin Jahedi, Yaroslava Nalivayko, and Andrés Bruhn. Spring: A high-resolution high-detail dataset and benchmark for scene flow, optical flow and stereo. In *CVPR*, 2023.

Nando Metzger, Rodrigo Caye Daudt, and Konrad Schindler. Guided depth super-resolution by deep anisotropic diffusion. In *CVPR*, 2023.

S. Mahdi H. Miangoleh, Sebastian Dille, Long Mai, Sylvain Paris, and Yagiz Aksoy. Boosting monocular depth estimation models to high-resolution via content-adaptive multi-resolution merging. In *CVPR*, 2021.

Simon Niklaus, Long Mai, Jimei Yang, and Feng Liu. 3D Ken Burns effect from a single image. In *SIGGRAPH*, 2019.

Maxime Oquab, Timothée Darcet, Théo Moutakanni, Huy Vo, Marc Szafraniec, Vasil Khalidov, Pierre Fernandez, Daniel Haziza, Francisco Massa, Alaaeldin El-Nouby, Mahmoud Assran, Nicolas Ballas, Wojciech Galuba, Russell Howes, Po-Yao Huang, Shang-Wen Li, Ishan Misra, Michael Rabbat, Vasu Sharma, Gabriel Synnaeve, Hu Xu, Hervé Jegou, Julien Mairal, Patrick Labatut, Armand Joulin, and Piotr Bojanowski. DINOv2: Learning robust visual features without supervision. *TMLR*, 2024.

Juewen Peng, Zhiguo Cao, Xianrui Luo, Hao Lu, Ke Xian, and Jianming Zhang. BokehMe: When neural rendering meets classical rendering. In *CVPR*, 2022a.

Zhiliang Peng, Li Dong, Hangbo Bao, Qixiang Ye, and Furu Wei. BEiT v2: Masked image modeling with vector-quantized visual tokenizers. *arXiv*, 2022b.

Andra Petrovai and Sergiu Nedevschi. Exploiting pseudo labels in a self-supervised learning framework for improved monocular depth estimation. In *CVPR*, 2022.

Luigi Piccinelli, Yung-Hsu Yang, Christos Sakaridis, Mattia Segù, Siyuan Li, Luc Van Gool, and Fisher Yu. UniDepth: Universal monocular metric depth estimation. In *CVPR*, 2024.

Xuebin Qin, Hang Dai, Xiaobin Hu, Deng-Ping Fan, Ling Shao, and Luc Van Gool. Highly accurate dichotomous image segmentation. In *ECCV*, 2022.

Alec Radford, Jong Wook Kim, Chris Hallacy, Aditya Ramesh, Gabriel Goh, Sandhini Agarwal, Girish Sastry, Amanda Askell, Pamela Mishkin, Jack Clark, Gretchen Krueger, and Ilya Sutskever. Learning transferable visual models from natural language supervision. *PMLR*, 2021.

Michaël Ramamonjisoa and Vincent Lepetit. SharpNet: Fast and accurate recovery of occluding contours in monocular depth estimation. In *ICCV Workshop*, 2019.

Michaël Ramamonjisoa, Yuming Du, and Vincent Lepetit. Predicting sharp and accurate occlusion boundaries in monocular depth estimation using displacement fields. In *CVPR*, 2020.

Pierluigi Zama Ramirez, Alex Costanzino, Fabio Tosi, Matteo Poggi, Samuele Salti, Stefano Mattoccia, and Luigi Di Stefano. Booster: A benchmark for depth from images of specular and transparent surfaces. *TPAMI*, 46(1), 2024.

René Ranftl, Alexey Bochkovskiy, and Vladlen Koltun. Vision transformers for dense prediction. In *ICCV*, 2021.

René Ranftl, Katrin Lasinger, David Hafner, Konrad Schindler, and Vladlen Koltun. Towards robust monocular depth estimation: Mixing datasets for zero-shot cross-dataset transfer. *TPAMI*, 44(3), 2022.

Mike Roberts, Jason Ramapuram, Anurag Ranjan, Atulit Kumar, Miguel Ángel Bautista, Nathan Paczan, Russ Webb, and Joshua M. Susskind. Hypersim: A photorealistic synthetic dataset for holistic indoor scene understanding. In *ICCV*, 2021.

Ashutosh Saxena, Min Sun, and Andrew Y. Ng. Make3D: Learning 3D scene structure from a single still image. *TPAMI*, 31(5), 2009.

Saurabh Saxena, Junhwa Hur, Charles Herrmann, Deqing Sun, and David J. Fleet. Zero-shot metric depth with a field-of-view conditioned diffusion model. *arXiv*, 2023.

Hanno Scharr, Stefan Körkel, and Bernd Jähne. Numerische Isotropieoptimierung von FIR-Filtern mittels Querglättung. In *DAGM-Symposium*, 1997.

Daniel Scharstein, Heiko Hirschmüller, York Kitajima, Greg Krathwohl, Nera Nesic, Xi Wang, and Porter Westling. High-resolution stereo datasets with subpixel-accurate ground truth. In *GCPR*, 2014.

Thomas Schöps, Johannes L. Schönberger, S. Galliani, Torsten Sattler, Konrad Schindler, Marc Pollefeys, and Andreas Geiger. A multi-view stereo benchmark with high-resolution images and multi-camera videos. In *CVPR*, 2017.

Meng-Li Shih, Shih-Yang Su, Johannes Kopf, and Jia-Bin Huang. 3D photography using context-aware layered depth inpainting. In *CVPR*, 2020.

Nathan Silberman, Derek Hoiem, Pushmeet Kohli, and Rob Fergus. Indoor segmentation and support inference from RGBD images. In *ECCV*, 2012.

Shuran Song, Samuel P. Lichtenberg, and Jianxiong Xiao. SUN RGB-D: A RGB-D scene understanding benchmark suite. In *CVPR*, 2015.

Jaime Spencer, Chris Russell, Simon Hadfield, and Richard Bowden. Deconstructing self-supervised monocular reconstruction: The design decisions that matter. *TMLR*, 2022.

Jaime Spencer, Simon Hadfield, Chris Russell, and Richard Bowden. Kick back & relax: Learning to reconstruct the world by watching SlowTV. In *ICCV*, 2023.

Jaime Spencer, Chris Russell, Simon Hadfield, and Richard Bowden. Kick back & relax++: Scaling beyond ground-truth depth with SlowTV & CribsTV. *arXiv*, 2024.

Deqing Sun, Daniel Vlasic, Charles Herrmann, Varun Jampani, Michael Krainin, Huiwen Chang, Ramin Zabih, William T. Freeman, and Ce Liu. AutoFlow: Learning a better training set for optical flow. In *CVPR*, 2021.

Quan Sun, Yuxin Fang, Ledell Wu, Xinlong Wang, and Yue Cao. EVA-CLIP: Improved training techniques for CLIP at scale. *arXiv*, 2023.

Mingxing Tan and Quoc Le. EfficientNetV2: Smaller models and faster training. In *ICML*, 2021.

Bart Thomee, David A Shamma, Gerald Friedland, Benjamin Elizalde, Karl Ni, Douglas Poland, Damian Borth, and Li-Jia Li. YFCC100M: The new data in multimedia research. *Communications of the ACM*, 59(2), 2016.

Fabio Tosi, Yiyi Liao, Carolin Schmitt, and Andreas Geiger. SMD-Nets: Stereo mixture density networks. In *CVPR*, 2021.

Hugo Touvron, Matthieu Cord, and Hervé Jégou. DeiT III: Revenge of the ViT. In *ECCV*, 2022.

Ashish Vaswani, Noam Shazeer, Niki Parmar, Jakob Uszkoreit, Llion Jones, Aidan N Gomez, Łukasz Kaiser, and Illia Polosukhin. Attention is all you need. In *NIPS*, 2017.

Qiang Wang, Shizhen Zheng, Qingsong Yan, Fei Deng, Kaiyong Zhao, and Xiaowen Chu. IRS: A large synthetic indoor robotics stereo dataset for disparity and surface normal estimation. *arXiv*, 2019.

Wenshan Wang, Delong Zhu, Xiangwei Wang, Yaoyu Hu, Yuheng Qiu, Chen Wang, Yafei Hu, Ashish Kapoor, and Sebastian A. Scherer. TartanAir: A dataset to push the limits of visual SLAM. In *IROS*, 2020.

Ross Wightman. Pytorch image models. `https://github.com/rwightman/pytorch-image-models`, 2019.

Sanghyun Woo, Shoubhik Debnath, Ronghang Hu, Xinlei Chen, Zhuang Liu, In So Kweon, and Saining Xie. ConvNeXt V2: Co-Designing and Scaling ConvNets With Masked Autoencoders. In *CVPR*, 2023.

Magnus Wrenninge and Jonas Unger. Synscapes: A photorealistic synthetic dataset for street scene parsing. *arXiv*, 2018.

Chunlong Xia, Xinliang Wang, Feng Lv, Xin Hao, and Yifeng Shi. ViT-CoMer: Vision transformer with convolutional multi-scale feature interaction for dense predictions. In *CVPR*, 2024.

Ke Xian, Chunhua Shen, Zhiguo Cao, Hao Lu, Yang Xiao, Ruibo Li, and Zhenbo Luo. Monocular relative depth perception with web stereo data supervision. In *CVPR*, 2018.

Ke Xian, Jianming Zhang, Oliver Wang, Long Mai, Zhe Lin, and Zhiguo Cao. Structure-guided ranking loss for single image depth prediction. In *CVPR*, 2020.

Enze Xie, Wenhai Wang, Zhiding Yu, Anima Anandkumar, José M. Álvarez, and Ping Luo. Seg-Former: Simple and efficient design for semantic segmentation with transformers. In *NeurIPS*, 2021.

Lihe Yang, Bingyi Kang, Zilong Huang, Xiaogang Xu, Jiashi Feng, and Hengshuang Zhao. Depth anything: Unleashing the power of large-scale unlabeled data. In *CVPR*, 2024a.

Lihe Yang, Bingyi Kang, Zilong Huang, Zhen Zhao, Xiaogang Xu, Jiashi Feng, and Hengshuang Zhao. Depth anything v2. In *NeurIPS*, 2024b.

Yao Yao, Zixin Luo, Shiwei Li, Jingyang Zhang, Yufan Ren, Lei Zhou, Tian Fang, and Long Quan. BlendedMVS: A large-scale dataset for generalized multi-view stereo networks. In *CVPR*, 2020.

Wei Yin, Jianming Zhang, Oliver Wang, Simon Niklaus, Long Mai, Simon Chen, and Chunhua Shen. Learning to recover 3D scene shape from a single image. In *CVPR*, 2021.

Wei Yin, Chi Zhang, Hao Chen, Zhipeng Cai, Gang Yu, Kaixuan Wang, Xiaozhi Chen, and Chunhua Shen. Metric3D: Towards zero-shot metric 3D prediction from a single image. In *ICCV*, 2023.

Weihao Yu, Chenyang Si, Pan Zhou, Mi Luo, Yichen Zhou, Jiashi Feng, Shuicheng Yan, and Xinchao Wang. Metaformer baselines for vision. *TPAMI*, 46(2), 2024.

Yuhui Yuan, Rao Fu, Lang Huang, Weihong Lin, Chao Zhang, Xilin Chen, and Jingdong Wang. HRFormer: High-resolution transformer for dense prediction. In *NeurIPS*, 2021.

Xiaohua Zhai, Basil Mustafa, Alexander Kolesnikov, and Lucas Beyer. Sigmoid loss for language image pre-training. In *ICCV*, 2023.

Chi Zhang, Wei Yin, Gang Yu, Zhibin Wang, Tao Chen, Bin Fu, Joey Tianyi Zhou, and Chunhua Shen. Robust geometry-preserving depth estimation using differentiable rendering. In *ICCV*, 2023a.

Lvmin Zhang, Anyi Rao, and Maneesh Agrawala. Adding conditional control to text-to-image diffusion models. In *ICCV*, 2023b.

Xuaner Zhang, Qifeng Chen, Ren Ng, and Vladlen Koltun. Zoom to learn, learn to zoom. In *CVPR*, 2019.

Zhiwei Zhong, Xianming Liu, Junjun Jiang, Debin Zhao, and Xiangyang Ji. Guided depth map super-resolution: A survey. *ACM Computing Surveys*, 2023.

Xizhou Zhu, Weijie Su, Lewei Lu, Bin Li, Xiaogang Wang, and Jifeng Dai. Deformable DETR: Deformable transformers for end-to-end object detection. In *ICLR*, 2021.

## SUPPLEMENTAL MATERIAL

In Section A, we provide additional results and experiments. Sec. A.1 presents further qualitative comparisons to baselines, Sec.A.2 presents a more detailed zero-shot evaluation, Sec.A.3 lists runtimes for all evaluated methods, and Sec.A.4 presents additional experiments on boundary accuracy. Section B showcases a selection of controlled experiments on Depth Pro that helped guide architectural choices (Sec.B.2, Sec.B.3, and Sec.B.7), training objective design (Sec.B.5), and curriculum training (Sec.B.6). In Section C, we provide additional implementation, training and evaluation details, including a complete summary of the datasets that were involved in this paper. Finally, Section D provides additional material on downstream applications.

## A  ADDITIONAL RESULTS

### A.1  QUALITATIVE RESULTS

We provide additional qualitative results of Depth Pro, Marigold (Ke et al., 2024), Metric3D v2 (Hu et al., 2024), and Depth Anything v2 (Yang et al., 2024b) on in-the-wild images from AM-2k (Li et al., 2022a), DIS-5k (Qin et al., 2022), and Unsplash[1] in Fig. 5, Fig. 6, and Fig. 7. Fine details are repeatedly missed by Metric3D v2 and Depth Anything v2. Marigold reproduces finer details than Metric3D v2 and Depth Anything v2 but commonly yields noisy predictions.

### A.2  ZERO-SHOT METRIC DEPTH

Expanding on the summary in Tab.1, we provide additional results for zero-shot metric depth estimation in Tab. 4. We report results on Booster (Ramirez et al., 2024), Middlebury (Scharstein et al., 2014), Sun-RGBD (Song et al., 2015), ETH3D (Schöps et al., 2017), nuScenes (Caesar et al., 2020), and Sintel (Butler et al., 2012). Our baselines include Depth Anything (Yang et al., 2024a) and Depth Anything v2 (Yang et al., 2024b), Metric3D (Yin et al., 2023) and Metric3D v2 (Hu et al., 2024), PatchFusion (Li et al., 2024a), UniDepth (Piccinelli et al., 2024), ZeroDepth (Guizilini et al., 2023), and ZoeDepth (Bhat et al., 2023). To preserve the zero-shot setting, we do not report results for models that were trained on the same dataset as the evaluation dataset. We report commonly used metrics in the depth estimation literature, namely $AbsRel$, $Log_{10}$ (Saxena et al., 2009), $\delta_1$, $\delta_2$ and $\delta_3$ scores (Ladicky et al., 2014), as well as point-cloud metrics (Spencer et al., 2022). Due to the high resolution of Booster images, we were not able to obtain point-cloud metrics in reasonable time.

---

[1] https://www.unsplash.com

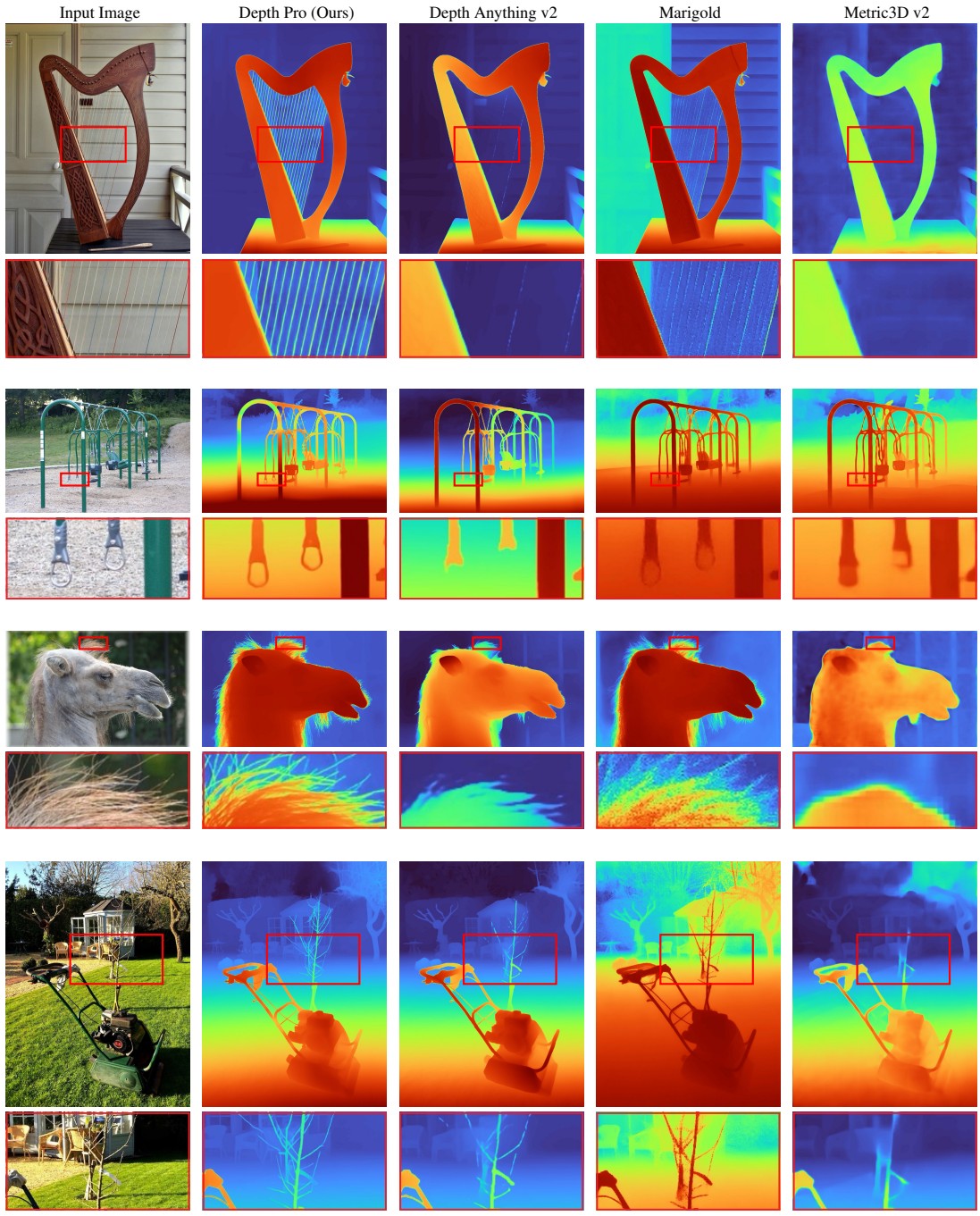

Figure 5: Zero-shot results of Depth Pro, Marigold (Ke et al., 2024), Metric3D v2 (Hu et al., 2024), and Depth Anything v2 (Yang et al., 2024b) on images from Unsplash (Li et al., 2022a), AM-2k (Li et al., 2022a), and DIS-5k (Qin et al., 2022).

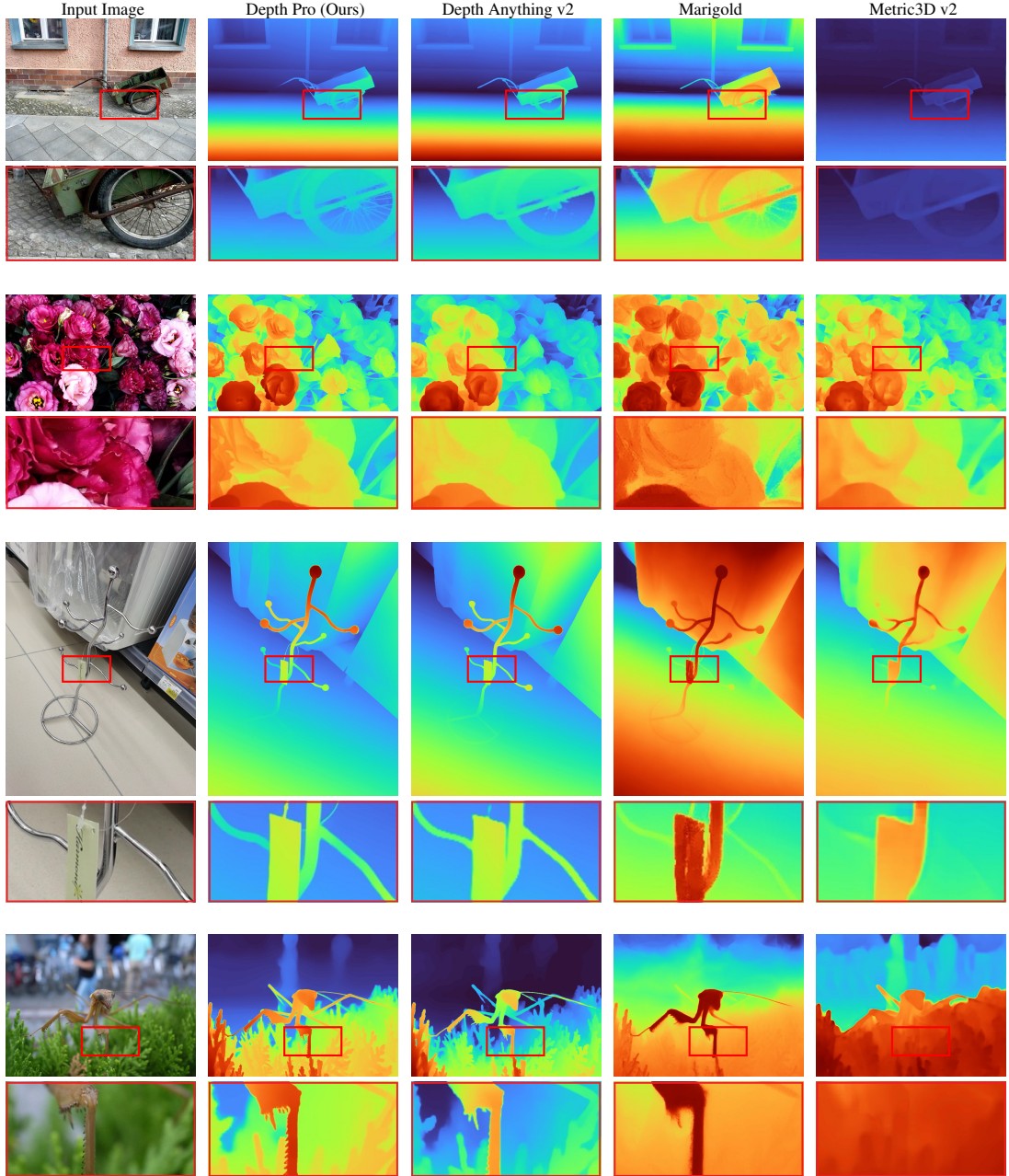

Figure 6: Zero-shot results of Depth Pro, Marigold (Ke et al., 2024), Metric3D v2 (Hu et al., 2024), and Depth Anything v2 (Yang et al., 2024b) on images from Unsplash (Li et al., 2022a), AM-2k (Li et al., 2022a), and DIS-5k (Qin et al., 2022).

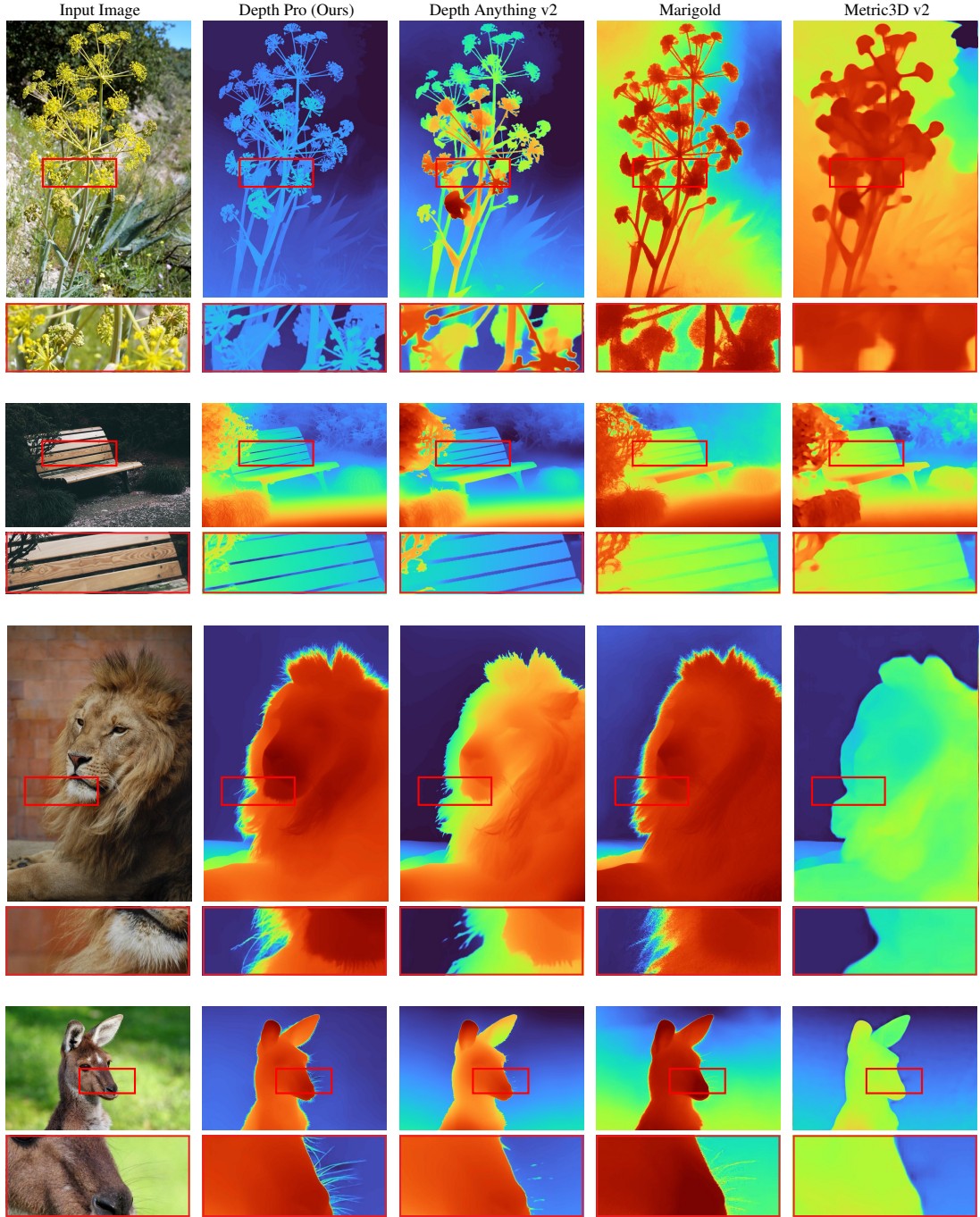

Figure 7: Zero-shot results of Depth Pro, Marigold (Ke et al., 2024), Metric3D v2 (Hu et al., 2024), and Depth Anything v2 (Yang et al., 2024b) on images from Unsplash (Li et al., 2022a), AM-2k (Li et al., 2022a), and DIS-5k (Qin et al., 2022).

Table 4: **Additional zero-shot metric depth evaluation.** We report additional metrics used in the depth estimation literature, namely $AbsRel$ (Ladicky et al., 2014), $Log_{10}$, $\delta_2$ and $\delta_3$ scores, as well as point-cloud metrics (Spencer et al., 2022) on Booster (Ramirez et al., 2024), Middlebury (Scharstein et al., 2014), Sun-RGBD (Song et al., 2015), ETH3D (Schöps et al., 2017), nuScenes (Caesar et al., 2020), and Sintel (Butler et al., 2012). For fair comparison, all reported results were reproduced in our environment.

| NuScenes | AbsRel↓ | Log₁₀↓ | δ₂↑ | δ₃↑ | SI-Log↓ | PC-CD↓ | PC-F↑ | PC-IoU↑ |
|---|---|---|---|---|---|---|---|---|
| DepthAnything (Yang et al., 2024a) | 0.453 | 0.151 | 73.876 | 90.301 | 28.153 | 24.146 | 0.007 | 0.004 |
| DepthAnything v2 (Yang et al., 2024b) | 0.614 | 0.326 | 31.837 | 47.265 | 29.737 | 37.516 | 0.008 | 0.004 |
| Metric3D (Yin et al., 2023) | 0.422 | 0.132 | 77.220 | 83.605 | 33.827 | 29.284 | 0.007 | 0.004 |
| Metric3D v2 (Hu et al., 2024) | 0.197 | 0.080 | **93.252** | 95.736 | 27.032 | 14.876 | 0.008 | 0.004 |
| PatchFusion (Li et al., 2024a) | 0.392 | 0.226 | 48.742 | 76.035 | 31.171 | 20.836 | 0.006 | 0.003 |
| UniDepth (Piccinelli et al., 2024) | 0.138 | 0.060 | 93.006 | 96.415 | 21.801 | 11.629 | 0.009 | 0.004 |
| ZeroDepth (Guizilini et al., 2023) | 0.237 | 0.121 | 82.596 | 89.908 | 30.703 | 23.348 | 0.007 | 0.004 |
| ZoeDepth (Bhat et al., 2023) | 0.498 | 0.182 | 64.947 | 82.704 | 31.501 | 39.183 | 0.006 | 0.003 |
| Depth Pro (Ours) | 0.287 | 0.164 | 73.836 | 84.252 | 29.548 | 22.480 | **0.010** | **0.005** |

| Sintel | AbsRel↓ | Log₁₀↓ | δ₂↑ | δ₃↑ | SI-Log↓ | PC-CD↓ | PC-F↑ | PC-IoU↑ |
|---|---|---|---|---|---|---|---|---|
| DepthAnything (Yang et al., 2024a) | 3.973 | 0.559 | 15.418 | 27.281 | 35.771 | 38.592 | 0.057 | 0.030 |
| DepthAnything v2 (Yang et al., 2024b) | 2.226 | 0.494 | 18.696 | 33.820 | 41.923 | 54.931 | 0.057 | 0.031 |
| Metric3D (Yin et al., 2023) | 1.733 | 0.387 | 32.375 | 44.793 | 48.605 | 45.858 | 0.056 | 0.031 |
| Metric3D v2 (Hu et al., 2024) | **0.370** | **0.216** | 62.915 | 76.866 | 25.312 | 34.790 | 0.091 | 0.051 |
| PatchFusion (Li et al., 2024a) | 0.617 | 0.391 | 35.515 | 51.443 | 36.806 | 44.615 | 0.077 | 0.045 |
| UniDepth (Piccinelli et al., 2024) | 0.869 | 0.301 | 35.722 | 57.256 | 42.837 | **32.338** | 0.098 | 0.057 |
| ZeroDepth (Guizilini et al., 2023) | 0.703 | 0.491 | 25.629 | 37.076 | 50.839 | 76.274 | 0.052 | 0.029 |
| ZoeDepth (Bhat et al., 2023) | 0.946 | 0.392 | 22.698 | 44.969 | 40.217 | 52.301 | 0.085 | 0.049 |
| Depth Pro (Ours) | 0.508 | 0.230 | 59.247 | 71.138 | 27.494 | 41.968 | **0.121** | **0.073** |

| Sun-RGBD | AbsRel↓ | Log₁₀↓ | δ₂↑ | δ₃↑ | SI-Log↓ | PC-CD↓ | PC-F↑ | PC-IoU↑ |
|---|---|---|---|---|---|---|---|---|
| DepthAnything (Yang et al., 2024a) | 0.114 | 0.053 | 98.811 | 99.770 | 8.038 | 0.034 | 0.160 | 0.090 |
| DepthAnything v2 (Yang et al., 2024b) | 0.182 | 0.070 | 97.645 | 99.462 | 8.390 | 0.045 | 0.169 | 0.096 |
| Metric3D (Yin et al., 2023) | 1.712 | 0.382 | 26.999 | 34.116 | 20.262 | 0.506 | 0.060 | 0.032 |
| Metric3D v2 (Hu et al., 2024) | 0.156 | 0.076 | 96.348 | 99.548 | 7.433 | 0.025 | 0.179 | 0.102 |
| PatchFusion (Li et al., 2024a) | 0.466 | 0.961 | 60.145 | 60.651 | 69.647 | 331.477 | 0.052 | 0.027 |
| UniDepth (Piccinelli et al., 2024) | **0.087** | **0.037** | **99.330** | **99.804** | **6.968** | **0.020** | **0.294** | **0.183** |
| ZoeDepth (Bhat et al., 2023) | 0.123 | 0.053 | 97.954 | 99.505 | 8.964 | 0.048 | 0.135 | 0.075 |
| Depth Pro (Ours) | 0.113 | 0.049 | 98.506 | 99.547 | 7.841 | 0.039 | 0.179 | 0.103 |

| ETH3D | AbsRel↓ | Log₁₀↓ | δ₂↑ | δ₃↑ | SI-Log↓ | PC-CD↓ | PC-F↑ | PC-IoU↑ |
|---|---|---|---|---|---|---|---|---|
| DepthAnything (Yang et al., 2024a) | 1.682 | 0.380 | 19.784 | 31.057 | 10.903 | 0.072 | 0.172 | 0.114 |
| DepthAnything v2 (Yang et al., 2024b) | 0.370 | 0.173 | 64.657 | 86.256 | 9.683 | 0.042 | 0.330 | 0.233 |
| Metric3D (Yin et al., 2023) | 0.859 | 0.240 | 49.291 | 57.573 | 14.541 | 0.072 | 0.303 | 0.219 |
| Metric3D v2 (Hu et al., 2024) | **0.124** | **0.053** | **99.553** | **99.900** | **6.197** | 0.083 | 0.466 | 0.358 |
| PatchFusion (Li et al., 2024a) | 0.256 | 0.106 | 88.378 | 97.306 | 11.023 | 0.042 | 0.209 | 0.135 |
| UniDepth (Piccinelli et al., 2024) | 0.457 | 0.186 | 57.670 | 81.483 | 7.729 | **0.031** | 0.409 | 0.305 |
| ZoeDepth (Bhat et al., 2023) | 0.500 | 0.176 | 64.452 | 81.434 | 13.250 | 0.078 | 0.127 | 0.082 |
| Depth Pro (Ours) | 0.327 | 0.193 | 61.309 | 71.228 | 10.170 | 0.094 | **0.487** | **0.398** |

| Middlebury | AbsRel↓ | Log₁₀↓ | δ₂↑ | δ₃↑ | SI-Log↓ | PC-CD↓ | PC-F↑ | PC-IoU↑ |
|---|---|---|---|---|---|---|---|---|
| DepthAnything (Yang et al., 2024a) | 0.273 | 0.149 | 69.619 | 86.060 | 12.420 | 0.102 | 0.103 | 0.055 |
| DepthAnything v2 (Yang et al., 2024b) | 0.262 | 0.141 | 72.074 | 90.549 | 9.639 | 0.063 | 0.127 | 0.069 |
| Metric3D (Yin et al., 2023) | 1.251 | 0.305 | 37.528 | 58.733 | 12.091 | 0.069 | 0.069 | 0.036 |
| Metric3D v2 (Hu et al., 2024) | 0.450 | 0.152 | 73.321 | 88.610 | **5.519** | **0.022** | 0.215 | 0.122 |
| PatchFusion (Li et al., 2024a) | 0.250 | 0.108 | 87.166 | 98.154 | 14.641 | 0.319 | 0.084 | 0.044 |
| UniDepth (Piccinelli et al., 2024) | 0.324 | 0.127 | 80.047 | **99.621** | 7.379 | 0.113 | **0.221** | **0.129** |
| ZeroDepth (Guizilini et al., 2023) | 0.377 | 0.179 | 67.060 | 78.952 | 14.482 | 0.232 | 0.052 | 0.027 |
| ZoeDepth (Bhat et al., 2023) | **0.214** | 0.115 | 77.683 | 90.860 | 10.448 | 0.069 | 0.114 | 0.062 |
| Depth Pro (Ours) | 0.251 | **0.089** | **93.169** | 96.401 | 8.610 | 0.107 | 0.161 | 0.091 |

| Booster | AbsRel↓ | Log₁₀↓ | δ₂↑ | δ₃↑ | SI-Log↓ |
|---|---|---|---|---|---|
| DepthAnything (Yang et al., 2024a) | 0.317 | 0.114 | **79.615** | 95.228 | 10.507 |
| DepthAnything v2 (Yang et al., 2024b) | 0.315 | 0.110 | 76.239 | 94.276 | 7.056 |
| Metric3D (Yin et al., 2023) | 1.332 | 0.346 | 13.073 | 33.975 | 10.631 |
| Metric3D v2 (Hu et al., 2024) | 0.417 | 0.140 | 75.783 | 92.833 | **3.932** |
| PatchFusion (Li et al., 2024a) | 0.719 | 0.213 | 49.387 | 72.892 | 14.128 |
| UniDepth (Piccinelli et al., 2024) | 0.500 | 0.166 | 60.904 | 89.213 | 7.436 |
| ZoeDepth (Bhat et al., 2023) | 0.610 | 0.195 | 52.655 | 75.508 | 10.551 |
| Depth Pro (Ours) | 0.336 | 0.118 | 79.429 | **96.524** | 4.616 |

## A.3 RUNTIME

To assess the latency of our approach in comparison to baselines, we test all approaches on images of varying sizes and report results in Tab. 5. We pick common image resolutions (VGA: 640×480, HD: 1920×1080, 4K: 4032×3024) and measure each method's average runtime for processing an image of the given size. All reported runtimes are reproduced in our environment and include preprocessing, eventual resizing (for methods operating at a fixed internal resolution), and inference of each model. We further report the parameter counts and flops (at HD resolution) for each method as measured with the fvcore package.

Among all approaches with a fixed output resolution, Depth Pro has the highest native output resolution, processing more than 3 times as many pixels as the next highest, Metric3D v2 (Hu et al., 2024). Yet Depth Pro has less than half the parameter count and requires only a third of the runtime compared to Metric3D v2.

The variable-resolution approaches (PatchFusion (Li et al., 2024a) and ZeroDepth (Guizilini et al., 2023)) have considerably larger runtime, with the faster model, ZeroDepth, taking almost 4 times as long as Depth Pro, even for small VGA images.

Table 5: **Model performance, measured on a V100-32G GPU.** We report runtimes in milliseconds (ms) on images of multiple sizes, as well as model parameter counts and flops. For fairness, the reported runtimes are reproduced in our environment. Entries are sorted by the native output resolution.

| Method | Parameter count | Flops$_{HD}\downarrow$ | Native output↑ resolution | | $t_{VGA}$ (ms) $\downarrow$ | $t_{HD}$ (ms) $\downarrow$ | $t_{4K}$ (ms) $\downarrow$ |
|---|---|---|---|---|---|---|---|
| DPT | 123M | - | 384 × 384 | = 0.15 MP | 33.2 | 30.6 | 27.8 |
| ZoeDepth | 340M | - | 384 × 512 | = 0.20 MP | 235.7 | 235.1 | 235.4 |
| DepthAnything v2 | 335M | 1827G | 518 × 518 | = 0.27 MP | 90.9 | 91.1 | 91.2 |
| UniDepth | 347M | 630G | 462 × 616 | = 0.28 MP | 178.5 | 183.0 | 198.1 |
| Metric3D | 203M | 477G | 480 × 1216 | = 0.58 MP | 217.9 | 263.8 | 398.1 |
| Marigold | 949M | - | 768 × 768 | = 0.59 MP | 5174.3 | 4433.6 | 4977.6 |
| Metric3D v2 | 1.378G | 6830G | 616 × 1064 | = 0.66 MP | 1299.6 | 1299.7 | 1390.2 |
| PatchFusion | 203M | - | Original (tile-based) | | 84012.0 | 84029.9 | 84453.9 |
| ZeroDepth | 233M | 10862G | Original | | 1344.3 | 8795.7 | 34992.2 |
| Depth Pro | 504M | 4370G | 1536 × 1536 | = 2.36 MP | 341.3 | 341.3 | 341.3 |

## A.4 BOUNDARY EXPERIMENTS

**Boundary metrics empirical study.** To illustrate how our boundary metrics work, we report additional qualitative edge metric results in Fig. 8. In particular, we show the occluding contours derived from the ground-truth and predicted depth, which illustrate how incorrect depth boundary predictions can impact the metric. Furthermore, to illustrate the behavior of the boundary precision and recall measurements under various image perturbations we also provide an empirical study in Fig. 9. We report both quantitative and qualitative results on samples from the UnrealStereo4K dataset (Tosi et al., 2021). Our results empirically demonstrate the correlation between erroneous depth edge predictions and low precision and recall values.

**Results on the iBims dataset (Koch et al., 2018).** We supplement our boundary evaluation by results on the iBims dataset, which is commonly used for evaluating depth boundaries. iBims consists of images of indoor scenes that have been laser-scanned. The images are at 640 × 480 resolution and have been supplemented with manually annotated occlusion boundary maps to facilitate evaluation. The iBims benchmark uses *Depth Directed Errors* (DDE), which evaluate overall metric depth accuracy, *Depth Boundary Errors* (DBE), which are similar in spirit to our proposed boundary metric but require manual annotation, and *Planar Errors*, which evaluate the accuracy of planes derived from the depth maps.

We find that Depth Pro is on par with the state of the art according to the DDE and PE metrics, and significantly outperforms all prior work according to the boundary metrics.

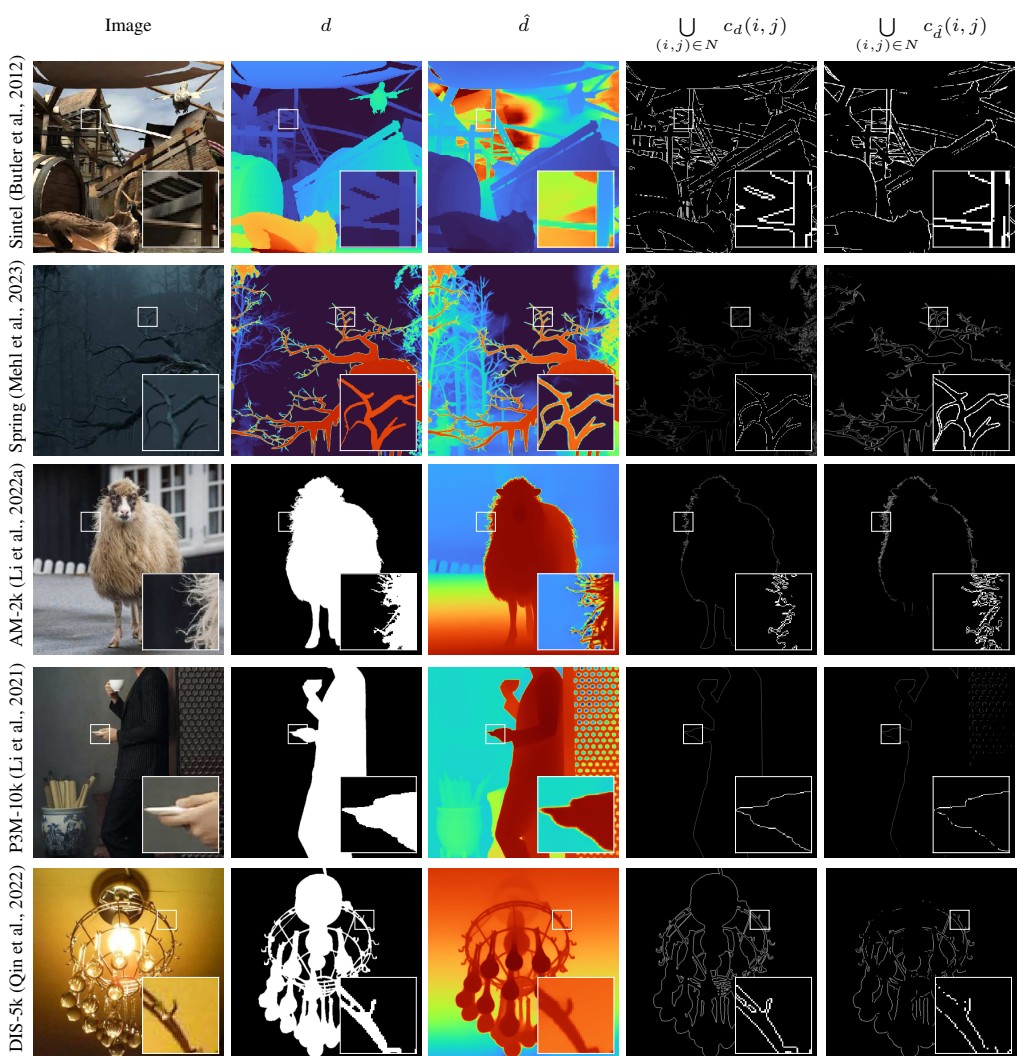

Figure 8: **Evaluation metrics for sharp boundaries.** We propose novel metrics to evaluate the sharpness of occlusion boundaries. The metrics can be computed on ground-truth depth maps (first two rows), and binary maps that can be derived from matting or segmentation datasets (subsequent rows). Each row shows a sample image, the ground truth for deriving occlusion boundaries, our prediction, ground-truth occluding contours, and occluding contours from the prediction. For these visualizations we set $t = 15$.

Table 6: **Zero-shot metric depth evalution on the iBims dataset (Koch et al., 2018).** We report the iBims-specific *Depth Directed Errors* (DDE), *Depth Boundary Errors* (DBE) and *Planar Errors* (PE). For fairness, all reported results were reproduced in our environment. Please see Sec. A.4

| Method | DDE (in %) | | | DBE (in px) | | PE (in m/°) | |
|---|---|---|---|---|---|---|---|
| | $\varepsilon_{\text{DDE}}^{0}\uparrow$ | $\varepsilon_{\text{DDE}}^{-}\downarrow$ | $\varepsilon_{\text{DDE}}^{+}\downarrow$ | $\varepsilon_{\text{DBE}}^{\text{acc}}\downarrow$ | $\varepsilon_{\text{DBE}}^{\text{comp}}\downarrow$ | $\varepsilon_{\text{PE}}^{\text{plan}}\downarrow$ | $\varepsilon_{\text{PE}}^{\text{orie}}\downarrow$ |
| DPT (Ranftl et al., 2021) | 58.744 | 41.255 | **0.000** | 3.580 | 39.372 | 0.138 | 31.837 |
| Metric3D (Yin et al., 2023) | 88.608 | 1.337 | 10.054 | 2.073 | 19.011 | 0.100 | 22.451 |
| Metric3D v2 (Hu et al., 2024) | 84.721 | 0.546 | 14.732 | 1.843 | 10.062 | **0.095** | 19.561 |
| ZoeDepth (Bhat et al., 2023) | 85.600 | 13.874 | 0.525 | 1.960 | 18.166 | 0.103 | 20.108 |
| Depth Anything (Yang et al., 2024a) | 88.951 | 10.741 | 0.308 | 2.081 | 19.172 | 0.106 | 20.680 |
| Depth Anything v2 (Yang et al., 2024b) | **91.773** | 1.619 | 6.607 | 1.959 | 8.350 | 0.095 | 19.406 |
| PatchFusion (Li et al., 2024a) | 85.765 | 12.602 | 1.633 | 1.711 | 20.722 | 0.117 | 23.926 |
| Marigold (Ke et al., 2024) | 58.738 | 41.261 | **0.000** | 1.855 | 12.742 | 0.168 | 33.734 |
| UniDepth (Piccinelli et al., 2024) | 73.020 | **0.041** | 26.939 | 1.999 | 14.234 | 0.098 | 19.114 |
| Depth Pro (Ours) | 89.725 | 1.809 | 8.464 | **1.680** | 10.138 | **0.095** | **18.776** |

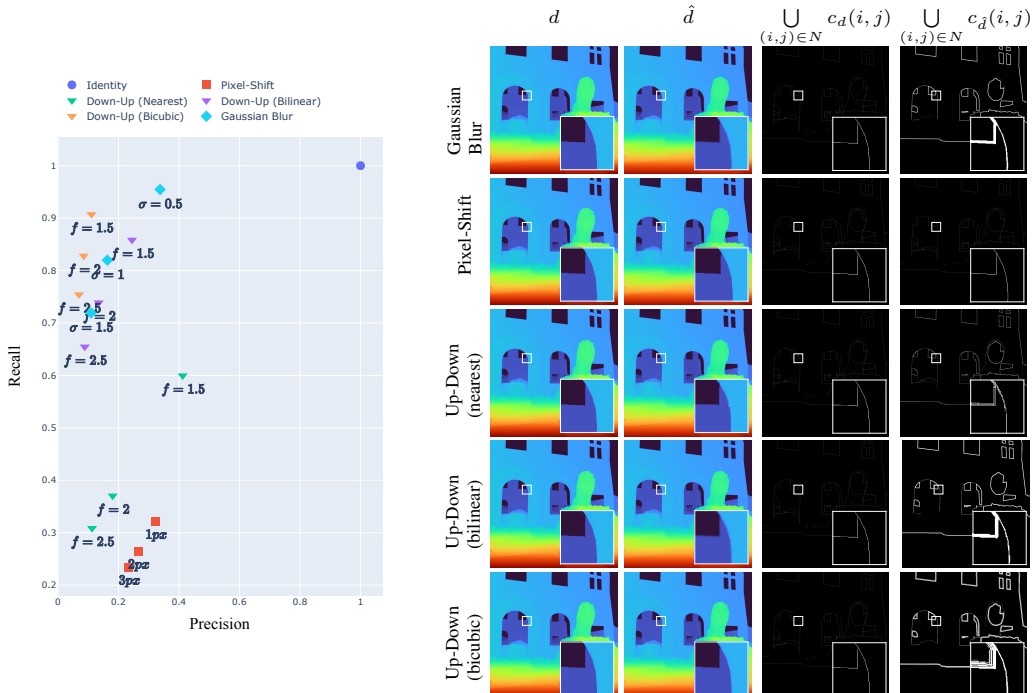

Figure 9: **Boundary evaluation metrics empirical study.** We demonstrate how various types of image perturbations impact our proposed edge metrics. We report quantitative and qualitative results for multiple ground-truth perturbations, such as simple image shifts, downsampling followed by upsamplings, and Gaussian blurring. We report both ground-truth and perturbed occluding contours, used to derive our $F1$ scores. Our results empirically demonstrate the correlation between erroneous depth edge predictions and low precision and recall values.

# B   CONTROLLED EXPERIMENTS

We conduct several controlled experiments to investigate the impact of various components and design decisions in Depth Pro. Specifically, we aim to assess the contribution of the native output resolution, key components in the network architecture (Sec. B.2), the depth representation, training objectives (Sec. B.5), training curriculum (Sec. B.6), and the focal length estimation head (Sec. B.7).

## B.1   NATIVE OUTPUT RESOLUTION

To assess the importance of the native output resolution, we conduct a simple experiment on UnrealStereo4K. We downsample the ground truth depth maps to several common resolutions found in the literature as well the output resolution of Depth Pro. We then upsample again to the original resolution and evaluate depth and boundary metrics. Results are listed in Tab. 7. We find that the native output resolution has a strong effect on boundary accuracy. More specifically, doubling the resolution may improve boundary accuracy by a factor of 3. It is important to note however, that these results represent an upper bound. The zero-shot comparison to the state of the art (Tab. 1 and 2) includes approaches that predict depth at the full input resolution, namely PatchFusion and ZeroDepth. Although PatchFusion and ZeroDepth predict at a higher resolution than e.g., Metric3D v2 or DepthAnything v2, they trail the lower resolution approaches in performance. We conclude that predicting at a high native output resolution is necessary but not sufficient for predicting accurate metric depth with sharp boundaries.

Table 7: **Native output resolution.** We evaluate the expected impact of the native output resolution on metric depth prediction and boundary accuracy. To that end, we bilinearly downsample the ground truth depth maps to several resolutions found in the literature, upsample again to the input resolution, and evaluate. The strong effect on boundary accuracy (3 fold increase per doubling of resolution) implies that predicting at a high native output resolution is a necessary but not necessarily sufficient condition for predicting accurate boundaries.

| Output resolution | Example | Approximate optimum | | |
| --- | --- | --- | --- | --- |
| | | Log10↓ | AbsRel↓ | F1↑ |
| 1536×1536 | Depth Pro | **0.019** | **0.004** | **0.311** |
| 768×768 | Marigold (Ke et al., 2024) | 0.048 | 0.010 | 0.131 |
| 518×518 | Depth Anything v2 (Yang et al., 2024a) | 0.084 | 0.016 | 0.065 |
| 384×384 | DPT (Ranftl et al., 2021) | 0.123 | 0.024 | 0.044 |

## B.2   NETWORK BACKBONE

To assess the effect of the network architecture, we begin by evaluating various candidate image encoder backbones within our network architecture. To assess their performance, we conduct a comparative analysis utilizing off-the-shelf models available from the TIMM library (Wightman, 2019). Using the pretrained weights, we train each backbone at $384 \times 384$ resolution across five RGB-D datasets (Keystone, HRWSI, RedWeb, TartanAir, and Hypersim) and evaluate their performance in terms of metric depth accuracy across multiple datasets, including Booster, Hypersim, Middlebury, and NYUv2, utilizing metrics such as $AbsRel$ for affine-invariant depth and $Log_{10}$ for metric depth in Tab. 8. We find that ViT-L DINOv2 (Oquab et al., 2024) outperforms all other backbones by a significant margin and conclude that the combination of backbone and pretraining strategy considerably affects downstream performance. Following this analysis, we pick ViT-L DINOv2 for our encoder backbones.

## B.3   HIGH-RESOLUTION ALTERNATIVES

We further evaluate alternative high-resolution 1536×1536 network structures and different pretrained weights (Tab. 9). To do this, we test generalization accuracy by training on a train split of some datasets and testing on a val or test split of other datasets, following the Stage 1 protocol for all models in accordance with Tab. 16 and Tab. 17. All ViT models use a patch size of 16×16. For weights pretrained with a patch size of 14×14 we apply bicubic interpolation to the weights of the

Table 8: **Comparison of image encoder backbones candidates.** We train each backbone at 384×384 resolution across five RGB-D datasets: Keystone, HRWSI, RedWeb, TartanAir, and Hypersim. To ensure fair comparison, we select backbone candidates with comparable computational complexity, measured in Flops using the fvcore library (Facebook Research, 2022), and an equivalent number of parameters. We identify ViT-L DINOv2 (Oquab et al., 2024) as the optimal choice for our image encoder backbone, given its superior depth accuracy performance.

| Backbone | Flops (G) | Params (M) | AbsRel $\downarrow$ | $Log_{10} \downarrow$ |
|---|---|---|---|---|
| ViT-L DINOv2-reg4 (Oquab et al., 2024) | 248 | 345 | **0.039** | 0.138 |
| ViT-L DINOv2 (Oquab et al., 2024) | 247 | 345 | 0.040 | **0.129** |
| ViT-L MAE (He et al., 2022) | 247 | 343 | 0.041 | 0.150 |
| ViT-L BeiTv2 (Peng et al., 2022b) | 242 | 336 | 0.042 | 0.134 |
| ViT-L BeiT (Bhat et al., 2023) | 259 | 336 | 0.048 | 0.147 |
| ViT-L SO400m-siglip (Zhai et al., 2023) | 311 | 471 | 0.051 | 0.174 |
| ViT-L CLIP-quickgelu (Radford et al., 2021) | 247 | 344 | 0.053 | 0.166 |
| ViT-L CLIP (Radford et al., 2021) | 247 | 345 | 0.057 | 0.156 |
| ViT-L (Dosovitskiy et al., 2021) | 247 | 345 | 0.061 | 0.163 |
| ConvNext-XXL (Liu et al., 2022b) | 514 | 867 | 0.075 | 0.216 |
| ViT-L DeiT-3 (Touvron et al., 2022) | 247 | 345 | 0.078 | 0.176 |
| ConvNext-L-mlp (Liu et al., 2022b) | 162 | 214 | 0.081 | 0.222 |
| ConvNextv2-H (Woo et al., 2023) | 405 | 680 | 0.085 | 0.242 |
| SegAnything ViT-L (Kirillov et al., 2023) | 245 | 330 | 0.087 | 0.311 |
| SWINv2-L (Liu et al., 2022a) | 177 | 212 | 0.091 | 0.240 |
| CAFormer-B36 (Yu et al., 2024) | 124 | 108 | 0.091 | 0.248 |
| EfficientViT-L3 (Liu et al., 2023) | – | – | 0.109 | 0.303 |

convolutional patch embedding layer and scale these weights inversely to the number of pixels (i.e., the weights are reduced by a factor of 1.3). All ViT models use resolution 1536×1536, for this we apply bicubic interpolation to positional embeddings prior to training. The Depth Pro approach in all cases uses ViT with resolution 384×384 and patch size 16×16 for both the patch encoder and the image encoder. SWINv2 and convolutional models are pretrained on ImageNet (Deng et al., 2009). Other models use different pretraining approaches described in their papers: CLIP (Radford et al., 2021), MAE (He et al., 2022), BeiTv2 (Peng et al., 2022b), and DINOv2 (Oquab et al., 2024). For the Segment Anything model we use publicly available pretrained weights, which were initialized using MAE pretraining (He et al., 2022) and subsequently trained for segmentation as described in their paper (Kirillov et al., 2023).

Table 9: **High-resolution alternatives.** Generalization accuracy of alternative high-resolution 1536×1536 models and different pretrained weights. All models are trained identically using Stage 1 in accordance with Tab. 16 and Tab. 17. Latency measured on a single GPU V100 with FP16 precision using batch=1. All ViT models use a patch size of 16×16. Depth Pro employs a ViT-L DINOv2 (Oquab et al., 2024) for the image and patch encoders.

| | Method | Latency, ms $\downarrow$ | Metric depth accuracy | | Boundary accuracy | |
|---|---|---|---|---|---|---|
| | | | NYUv2 $\delta_1 \uparrow$ | iBims $\delta_1 \uparrow$ | iBims F1 $\uparrow$ | DIS R $\uparrow$ |
| Conv. | EfficientNetV2-XL (Tan & Le, 2021) | 118 | 4.4 | 7.0 | 0.005 | 0.000 |
| | ConvNext-XXL (Liu et al., 2022b) | 304 | 68.0 | 38.3 | 0.134 | 0.031 |
| | ConvNextv2-H (Woo et al., 2023) | 287 | 70.0 | 56.6 | 0.131 | 0.044 |
| Trans. | S. Anything (Kirillov et al., 2023) (ViT-L) | 349 | 53.2 | 38.9 | 0.140 | 0.051 |
| | S. Anything (Kirillov et al., 2023) (ViT-H) | 365 | 51.7 | 41.1 | 0.146 | 0.050 |
| | SWINv2-L (Liu et al., 2022a) (window=24) | 272 | 58.4 | 33.1 | 0.117 | 0.028 |
| ViT | ViT-L CLIP (Radford et al., 2021) | 384 | 92.2 | 81.9 | 0.157 | 0.052 |
| | ViT-L BeiTv2 (Peng et al., 2022b) | OOM | 90.4 | 86.5 | 0.149 | 0.042 |
| | ViT-L MAE (He et al., 2022) | 390 | 92.7 | 84.7 | 0.163 | 0.065 |
| | ViT-L DINOv2 (Oquab et al., 2024) | 392 | **96.5** | 90.3 | 0.161 | 0.065 |
| | Depth Pro | 341 | 96.1 | **91.3** | **0.177** | **0.080** |

We find that the presented Depth Pro approach is faster and more accurate for object boundaries than the plain ViT, with comparable metric depth accuracy. In comparison to other transformer-based and

convolutional models, Depth Pro has comparable latency, several times lower metric depth error, and several times higher recall accuracy for object boundaries.

Importantly, our proposed architecture performs significantly better than straight-forward scaling up the ViT architecture with DINOv2 pretraining. On DIS5K for instance, our architecture improves the boundary recall by relative 23% over DINOv2.

## B.4 DEPTH REPRESENTATION

To assess the effect of the predicted depth representation, we train a ViT encoder and DPT decoder on Hypersim to predict inverse depth, log-depth, or depth. All configurations in this experiment are supervised with a only mean absolute error. Tab. 10 lists the $delta_1$ error computed over several depth ranges. We find that supervising the network on depth directly leads to worse results than log-depth or inverse depth. Overall, predicting inverse depth works best, with the largest difference in the regions close to the camera. This makes inverse depth the representation of choice for downstream tasks like novel view synthesis, which benefit particularly from higher accuracy close to the camera.

Table 10: **Depth representation.** Optimizing for inverse-depth yields the most accurate predictions near the camera, which is particularly important for novel-view synthesis applications.

| Training objective | Hypersim $\delta_1 \uparrow$ | | | | | |
|---|---|---|---|---|---|---|
| | 0-1m | 1-2m | 2-4m | 4-8m | 8-16m | >16m |
| Inverse-depth | **0.730** | **0.833** | **0.896** | **0.921** | **0.922** | **0.922** |
| Log-depth | 0.700 | 0.807 | 0.892 | 0.919 | 0.920 | 0.920 |
| Depth | 0.657 | 0.716 | 0.819 | 0.853 | 0.850 | 0.850 |

## B.5 TRAINING OBJECTIVES

To assess the efficacy of our training curriculum, we compare it to alternative training schedules. We first examine the different stages individually and then compare full curricula.

Table 11: **Comparison of stage 1 training objectives.** 1A only applies the $\mathcal{L}_{MAE}$ to metric, and the $\mathcal{L}_{SSI\text{-}MAE}$ to non-metric datasets. 1D additionally minimizes gradients on all datasets. 1B minimizes gradients only on synthetic datasets. We use 1C, which minimizes gradients with a scale-and-shift-invariant $\mathcal{L}_{SSI\text{-}MAGE}$ loss on all synthetic datasets irrespective of whether they are metric.

| Cond. | HRWSI | | | Hypersim | | | Apolloscape | | |
|---|---|---|---|---|---|---|---|---|---|
| | AbsRel$\downarrow$ | $\delta_1 \uparrow$ | Log$_{10}\downarrow$ | AbsRel$\downarrow$ | $\delta_1 \uparrow$ | F1$\uparrow$ | Log$_{10}\downarrow$ | AbsRel$\downarrow$ | $\delta_1 \uparrow$ |
| 1A | 0.166 | 82.1 | 0.083 | 0.259 | 75.4 | 0.221 | 0.156 | 0.339 | 45.6 |
| 1D | **0.138** | **85.1** | 0.077 | 0.246 | 78.4 | 0.391 | 0.128 | 0.424 | 60.6 |
| 1B | 0.156 | 83.3 | 0.078 | 0.249 | 77.3 | 0.388 | 0.152 | 0.300 | 47.3 |
| 1C | 0.150 | 83.7 | **0.074** | **0.235** | **79.9** | **0.442** | **0.084** | **0.235** | **75.6** |

**Stage 1 training objectives.** We first evaluate loss combinations for the first stage and report results in Tab. 11. Condition 1A only applies a mean absolute error loss to all datasets. For non-metric datasets, we use the scale-and-shift-invariant version. Condition 1B adds gradient losses to all synthetic datasets. We again use the scale-and-shift-invariant version for non-metric datasets. Following our observations from Sec. 3, we propose to apply an appropriate mean absolute error loss as in other conditions depending on a dataset being metric, but apply a scale-and-shift-invariant gradient loss irrespective of a dataset being metric or not (C). We find that loss combinations minimizing gradients (1B, 1C, 1D) consistently outperform just applying an absolute error (1A). Besides improving relative and metric depth estimates, they strongly improve boundary metrics, here up to a factor of 2. Interestingly, minimizing gradients on all datasets outperforms minimizing gradients on just synthetic data. This suggests that the added diversity from real-world datasets more than balances out their potentially noisy ground truth, even for minimizing gradients, which emphasize the noise. The

best performance however, is achieved by applying a scale-and-shift-invariant gradient loss on the synthetic datasets (1C). We found that this setting improves convergence and overall performs best.

Table 12: **Comparison of stage 2 training objectives.** We evaluate the efficacy of derivative-based losses for sharpening boundaries. Employing first- and second-order derivative losses (2A) yields the best results on balance as indicated by the average rank over metrics. More details in the text.

| Condition | $\mathcal{L}_{MSE}$ | $\mathcal{L}_{MAGE}$ | $\mathcal{L}_{MSGE}$ | $\mathcal{L}_{MALE}$ | HRWSI | | Hypersim | | | | Apolloscape | | |
|---|---|---|---|---|---|---|---|---|---|---|---|---|---|
| | | | | | AbsRel↓ | $\delta_1\uparrow$ | Log$_{10}$↓ | AbsRel↓ | $\delta_1\uparrow$ | F1↑ | Log$_{10}$↓ | AbsRel↓ | $\delta_1\uparrow$ |
| 2A | ✓ | ✓ | ✓ | ✓ | 0.149 | 83.6 | **0.072** | 0.235 | **81.3** | 0.465 | 0.092 | 0.303 | 72.9 |
| 2B | ✓ | ✓ | ✓ | | **0.148** | **83.7** | **0.072** | **0.230** | 81.0 | 0.463 | 0.092 | **0.299** | 73.1 |
| 2C | ✓ | ✓ | | | 0.150 | **83.7** | **0.072** | 0.235 | 80.8 | **0.468** | **0.091** | 0.300 | 73.2 |
| 2D | ✓ | | | | 0.150 | 83.4 | 0.074 | 0.239 | 79.8 | 0.461 | 0.096 | 0.349 | 72.8 |
| 2E | | | | | 0.159 | 82.7 | 0.074 | 0.242 | 80.6 | 0.459 | 0.096 | 0.346 | **73.3** |

**Stage 2 training objectives.** The second stage of our training curriculum focuses on sharpening depth boundaries while retaining high metric depth accuracy. To that end, we only employ synthetic datasets due to their high quality ground truth. The obvious strategy for sharpening predictions is the application of gradient losses. We evaluate our combination of multi-scale derivative-based losses in an ablation study. Condition 2A uses all of the losses, namely $\mathcal{L}_{MAE}$, $\mathcal{L}_{MSE}$, $\mathcal{L}_{MAGE}$, $\mathcal{L}_{MALE}$, and $\mathcal{L}_{MSGE}$. See Tab. 12. 2B removes the second-order loss $\mathcal{L}_{MALE}$. 2C further removes the squared first order losses $\mathcal{L}_{MSGE}$. 2D removes all derivative-based losses. 2E applies the $\mathcal{L}_{MAE}$ to all datasets. Removing $\mathcal{L}_{MALE}$ improves results on Apolloscape. Our combination of 0th- to 2nd-order derivative losses (2A) performs best across metrics and datasets in aggregate (e.g., in terms of the average rank across metrics).

## B.6 FULL CURRICULA

Table 13: **Comparison of full curricula.** We evaluate our curriculum (3A) against single stage training (3B), and pretraining on synthetic and fine-tuning on real data (3C).

| Cond. | HRWSI | | Hypersim | | | Apolloscape | | |
|---|---|---|---|---|---|---|---|---|
| | AbsRel↓ | $\delta_1\uparrow$ | Log$_{10}$↓ | AbsRel↓ | $\delta_1\uparrow$ | F1↑ | Log$_{10}$↓ | AbsRel↓ | $\delta_1\uparrow$ |
| 3A (Ours) | 0.149 | 83.6 | **0.072** | **0.235** | **81.3** | 0.465 | **0.092** | 0.303 | **72.9** |
| 3B | **0.148** | **83.9** | 0.073 | 0.245 | **81.3** | **0.478** | 0.095 | **0.292** | 72.1 |
| 3C | 0.153 | 83.6 | 0.166 | 0.386 | 37.1 | 0.095 | 0.586 | 0.712 | 0.5 |

We assess the efficacy of our complete training curriculum in comparison to alternatives. Condition 3A represents our two-stage curriculum. Condition 3B trains in a single stage and applies all the second-stage gradient losses throughout the whole training. Condition 3C reverses our two stages and represents the established strategy of pretraining on synthetic data first and fine-tuning with real-world data. We find that training with a single stage is a reasonable default strategy and works much better than first training on synthetic data and then fine-tuning on real data. Our proposed strategy however, yields further improvements on metric depth with slightly worse boundary accuracy.

## B.7 FOCAL LENGTH ESTIMATION

**Additional analysis of zero-shot focal length estimation accuracy.** In Fig. 10, we present a more comprehensive analysis of our focal length predictor's performance compared to baseline models. To that end, we plot the percentage of samples below a certain absolute relative error for each method and dataset in our zero-shot evaluation set up. Depth Pro outperforms all approaches on all datasets.

**Controlled evaluation of network structures.** We evaluate a number of choices for the focal length estimation head and report results in Tab. 14. The models are evaluated on 500 images randomly sampled from Flickr (Thomee et al., 2016). As the first condition, we extract features from the frozen image encoder trained for depth estimation and add a small convolutional head. As the second

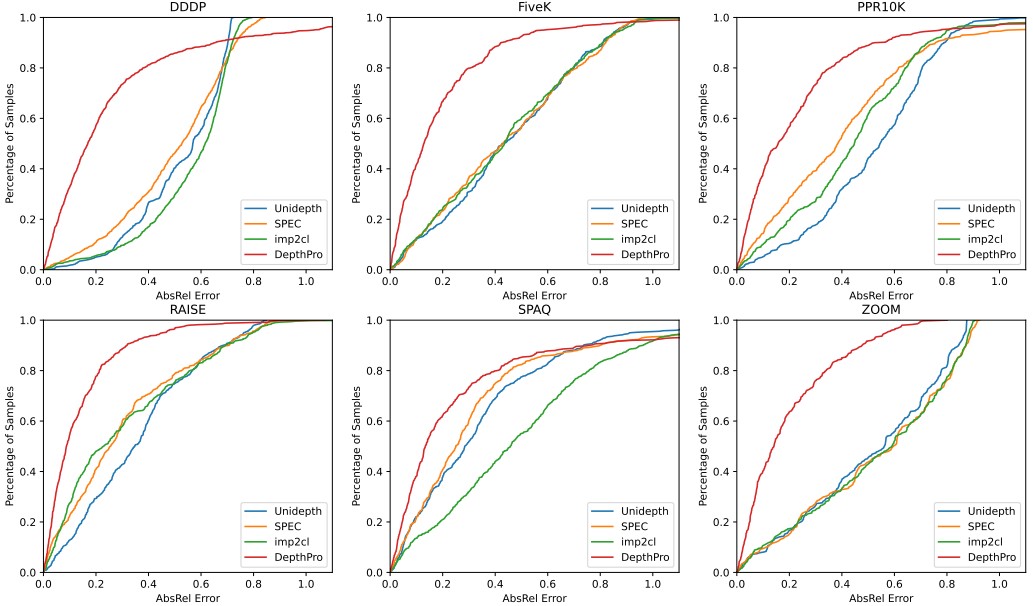

Figure 10: **Evaluation of focal length estimation.** Each plot compares a number of methods on a given dataset. The $x$ axis represents the AbsRel error and the $y$ axis represents the percentage of samples whose error is below that magnitude.

Table 14: Controlled experiment on focal length estimation. We evaluate several variants of a focal length branch and find that the combination of a separate image encoder trained for focal length estimation and a frozen image encoder for depth estimation perform best.

| Architecture | $\delta_{25\%} \uparrow$ |
|---|---|
| Encoder for depth only | 60.0 |
| Encoder for focal length only | 74.4 |
| Encoder for depth and refinement network | 63.6 |
| Parallel encoders for depth and focal length | **78.2** |

condition, we train a separate ViT-based image encoder (Dosovitskiy et al., 2021). As the third condition, we train a ViT-based encoder on extracted features from the frozen image encoder for depth estimation. The final condition represents our chosen architecture depicted in Fig. 3, which utilizes frozen features from the depth network and task-specific features from a separate ViT image encoder in parallel.

We observe that refining depth features performs on par with just using the frozen depth features, suggesting that adding more computation on top of the frozen DPT features in addition to our small convolutional head does not provide extra benefits despite the increased computation. Training a separate image encoder from scratch improves performance by 14.6 percentage points, which indicates that accurate focal length prediction requires extra task-specific knowledge in addition to depth information. Furthermore, the two encoders in parallel outperform using just a single image encoder for focal length prediction, which highlights the importance of features from the pretrained depth network for obtaining a high-performing focal length estimator.

## C  IMPLEMENTATION, TRAINING AND EVALUATION DETAILS

In this section we provide additional details on the datasets used for training and evaluation, hyperparameter settings for our method, and details on the evaluation setup.

### C.1  MERGE OPERATION

We merge overlapping feature patches to feature maps by generating a Voronoi partition of the desired feature map. To generate the partition, we use the patch centers as seeds and obtain a Voronoi cell per feature patch. The area of the patch covered by the Voronoi cell is copied to the feature map, the remaining area discarded. By overlapping patches we ensure that the receptive field of the patch encoder partially covers neighboring patches. Fig. 11 illustrates the approach for merging $3 \times 3$ patches into the *Feature 4* feature map.

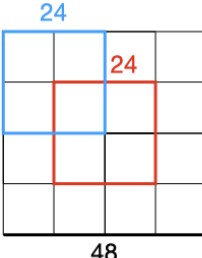 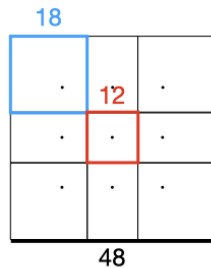

Figure 11: **Merge operation.** To merge overlapping feature patches (left), we generate a Voronoi partition of the feature map using the patch centers as generators (right). Red and blue rectangles exemplify the original feature patch on the left and the fraction to be retained on the right. The patch centers are indicated by dots on the right, numbers denote side lengths of the feature patches and the feature map.

### C.2  FOCAL LENGTH HEAD

The focal length head consists of a three-layer convolutional network with kernel sizes 3,3,6, and strides 2,2,1. Each layer halves the channel dimension and is followed by a rectified linear unit. The first layer starts from 128 channels and the last reduces channels to a single focal length value per image.

### C.3  DATASETS

Tab. 15 provides a comprehensive summary of the datasets utilized in our study, detailing their respective licenses and specifying their roles (e.g., training or testing).

Table 15: Datasets used in this work.

| Dataset | URL | License | Usage |
|---|---|---|---|
| 3D Ken Burns (Niklaus et al., 2019) | https://github.com/sniklaus/3d-ken-burns | CC-BY-NC-SA 4.0 | Train |
| AM-2K (Li et al., 2022a) | https://github.com/JizhiziLi/GFM | Custom | Testing |
| Apolloscape (Huang et al., 2020) | https://apolloscape.auto/ | Custom | Val |
| ARKitScenes (Dehghan et al., 2021) | https://github.com/apple/ARKitScenes | Custom | Train |
| Bedlam (Black et al., 2023) | https://bedlam.is.tue.mpg.de/#data | Custom | Train |
| BlendedMVG (Yao et al., 2020) | https://github.com/YoYo000/BlendedMVS | CC BY 4.0 | Train |
| Booster (Ramirez et al., 2024) | https://cvlab-unibo.github.io/booster-web/ | CC BY NC 4.0 | Test |
| DDAD (Guizilini et al., 2020) | https://github.com/TRI-ML/DDAD | CC-BY-NC-SA 4.0 | Testing |
| DIML (indoor) (Kim et al., 2016) | https://dimlrgbd.github.io/ | Custom | Train |
| DIS5K (Qin et al., 2022) | https://xuebinqin.github.io/dis/index.html | Custom | Test |
| DPDD (Abuolaim & Brown, 2020) | https://github.com/Abdullah-Ab...pixel | MIT | Testing |
| Dynamic Replica (Karaev et al., 2023) | https://github.com/facebookres...stereo | CC BY-NC 4.0 | Train |
| EDEN (Le et al., 2021) | https://lhoangan.github.io/eden/ | Custom | Train |
| ETH3D (Schöps et al., 2017) | https://www.eth3d.net/ | CC-BY-NC-SA 4.0 | Testing |
| FiveK (Bychkovsky et al., 2011) | https://data.csail.mit.edu/graphics/fivek | Custom | Testing |
| HRWSI (Xian et al., 2020) | https://kexianhust.github....Ranking-Loss/ | Custom | Train, Val |
| Hypersim (Roberts et al., 2021) | https://github.com/apple/ml-hypersim | Custom | Train, Val |
| iBims (Koch et al., 2018) | https://www.asg.ed.tum.de/lmf/ibims1/ | Custom | Test |
| IRS (Wang et al., 2019) | https://github.com/HKBU-HPML/IRS | Custom | Train |
| KITTI (Geiger et al., 2013) | https://www.cvlibs.net/datasets/kitti/ | CC-BY-NC-SA 3.0 | Testing |
| Middlebury (Scharstein et al., 2014) | https://vision.middlebury.edu/stereo/data/ | Custom | Testing |
| MVS-Synth (Huang et al., 2018) | https://phuang17....mvs-synth.html | Custom | Train |
| NYUv2 (Silberman et al., 2012) | https://cs.nyu.edu/...v2.html | Custom | Testing |
| nuScenes (Caesar et al., 2020) | https://www.nuscenes.org/ | Custom | Testing |
| P3M-10k (Li et al., 2021) | https://github.com/JizhiziLi/P3M | Custom | Testing |
| PPR10K (Liang et al., 2021) | https://github.com/csjliang/PPR10K | Apache 2.0 | Testing |
| RAISE (Dang-Nguyen et al., 2015) | http://loki...download.html | Custom | Testing |
| ReDWeb (Xian et al., 2018) | https://sites.google.com/site/redwebcvpr18/ | Custom | Train |
| SAILVOS3D (Hu et al., 2021) | https://sailvos.web.illin...index.html | Custom | Train |
| ScanNet (Dai et al., 2017) | http://www.scan-net.org/ | Custom | Train |
| Sintel (Butler et al., 2012) | http://sintel.is.tue.mpg.de/ | Custom | Testing |
| SmartPortraits (Kornilova et al., 2022) | https://mobile...SmartPortraits/ | Custom | Train |
| SPAQ (Fang et al., 2020) | https://github.com/h4nwei/SPAQ | Custom | Testing |
| Spring (Mehl et al., 2023) | https://spring-benchmark.org/ | CC BY 4.0 | Testing |
| Sun-RGBD (Song et al., 2015) | https://rgbd.cs.princeton.edu/ | Custom | Testing |
| Synscapes (Wrenninge & Unger, 2018) | https://synscapes.on.liu.se/ | Custom | Train |
| TartanAir (Wang et al., 2020) | https://theairlab.org/tartanair-dataset/ | CC BY 4.0 | Train |
| UASOL (Bauer et al., 2019) | https://osf.io/64532/ | CC BY 4.0 | Train |
| UnrealStereo4K (Tosi et al., 2021) | https://github.com/fabiotosi92/SMD-Nets | Custom | Train |
| Unsplash | https://unsplash.com/data | Custom | Testing |
| UrbanSyn (Gómez et al., 2023) | https://www.urbansyn.org/ | CC BY-SA 4.0 | Train |
| VirtualKITTI2 (Gaidon et al., 2016) | https://europe.naverlabs.com...-worlds/ | CC BY-NC-SA 3.0 | Train |
| ZOOM (Zhang et al., 2019) | https://github.com/ceciliav...inference | - | Testing |

Table 16: Training hyperparameters.

|  | Stage 1 | Stage 2 |
|---|---|---|
| Epochs | 250 | 100 |
| Epoch length | 72000 | |
| Schedule | 1 % warmup, 80 % constant LR, 19 % ×0.1 LR | |
| LR for Encoder | 1.28e-5 | |
| LR for Decoder | 1.28e-4 | |
| Batch size | 128 | |
| Optimizer | Adam | |
| Weight decay | 0 | |
| Clip gradient norm | 0.2 | |
| Pretrained LayerNorm | Frozen | |
| Random color change probability | 75 % | |
| Random blur probability | 30 % | |
| Center crop probability for FOV-augmentation | 50 % | |
| Metric depth normalization | CSTM-label (Yin et al., 2023) | |
| Number of channels for Decoder | 256 | |
| Resolution | 1536×1536 | |
| DepthPro model structure: | | |
| Image-Encoder resolution | 384×384 | |
| Patch-Encoder resolution | 384×384 | |
| Number of 384×384 patches in DepthPro | 35 | |
| Intersection of 384×384 patches in DepthPro | 25 % | |

Table 17: Training loss functions for different datasets and stages.

| Loss function | Datasets |
|---|---|
| **Stage 1** | |
| MAE
SSI-MAGE | Hypersim, Tartanair, Synscapes, Urbansyn, Dynamic Replica, Bedlam, IRS, Virtual Kitti2, Sail-vos3d |
| MAE (trimmed = 20 %) | ARKitScenes, Diml Indoor, Scannet, Smart Portraits |
| SSI-MAE
SSI-MAGE | UnrealStereo4k, 3D Ken Burns, Eden, MVS Synth |
| SSI-MAE (trimmed = 20 %) | HRWSI, BlendedMVG |
| **Stage 2** | |
| MAE, MSE, MAGE, MALE, MSGE | Hypersim, Tartanair, Synscapes, Urbansyn, Dynamic Replica, Bedlam, IRS, Virtual Kitti2, Sail-vos3d |

## C.4 TRAINING HYPERPARAMETERS

We specify the training hyperparameters in Tab. 16 and Tab. 17.

## C.5 BASELINES

Below we provide further details on the setup of the baselines.

**DepthAnything.** Depth Anything v1 and v2 each released a general model for *relative* depth, but their *metric* depth models are tailored to specific domains (indoor vs. outdoor). For the metric depth

Table 18: **Dataset evaluation setup.** For each metric depth dataset in our evaluation, we report the range of valid depth values, number of samples, and resolution of ground truth depth maps. Due to the large size of the validation set (approximately 35K samples), we used a randomly sampled subset of NuScenes.

| Dataset | Minimum distance (m) | Maximum distance (m) | Number of Samples | Depth Resolution (px) |
|---|---|---|---|---|
| Booster | 0.001 | 10 | 228 | $3008 \times 4112$ |
| ETH3D | 0.100 | 200 | 454 | $4032 \times 6048$ |
| iBims | 0.100 | 10 | 100 | $480 \times 640$ |
| Middlebury | 0.001 | 10 | 15 | $1988 \times 2952$ |
| NuScenes | 0.001 | 80 | 881 | $900 \times 1600$ |
| Sintel | 0.010 | 80 | 1064 | $436 \times 1024$ |
| Sun-RGBD | 0.001 | 10 | 5050 | $530 \times 730$ |

evaluation, we match these models to datasets according to their domain, and for datasets containing both indoor and outdoor images, we select the model with the best performance. For qualitative results and the (scale and shift invariant) zero-shot boundary evaluation, we employ the relative depth models, since they yield better qualitative results and sharper boundaries than the metric models.

**Metric3D.** For Metric3D v1 and v2, we found that the crop size parameter strongly affects metric scale accuracy. In fact, using a fixed crop size consistently yielded very poor results on at least one metric dataset. In order to obtain acceptable results, we used different crop sizes for indoor (512, 1088) and outdoor (512, 992) datasets. As in the case of Depth Anything, we mark these results in gray to indicate that they are not strictly zero-shot. For Metric 3D v2, we use the largest ('giant') model.

**UniDepth.** For UniDepth, we use the *ViT-L* version, which performs best on average among the UniDepth variants.

**ZoeDepth.** We use the model finetuned on both indoor and outdoor data (denoted *ZoeD_NK*).

### C.6 EVALUATION SETUP

In evaluating our approach and baselines, we found the range of valid depth values, the depth map resolution used for computing metrics, the resizing approach used for matching the resolution of the ground truth depth maps, and the choice of intrinsics to affect results, sometimes strongly. This is why we made an effort to set up and evaluate each baseline in the same fair evaluation setup, which we detail below.

Tab. 18 lists our evaluation datasets, the range of depth values used for evaluation, the number of samples, and the resolution of the ground truth depth maps. In case a method predicted depth maps at a different resolution, we resized predictions bilinearly to match the ground truth resolution.

Since several factors outlined above can affect the reported accuracy of a method, few baselines report sufficient detail on their evaluation setup, and the exact evaluation setups may differ across baselines, it is generally impossible to exactly reproduce reported results while guaranteeing fairness. We prioritized fair comparison and tried to evaluate all baselines in the same environment. We were able to match most reported results, with the following three notable differences. ZeroDepth reported better results on nuScenes, which we attribute to the use of a different validation set in their evaluation. UniDepth reported different results on ETH3D, which we attribute to the handling of raw images; specifically, in our setup, we use the raw images without any post-processing, and take the intrinsics from the accompanying EXIF data; we believe this best adheres to the zero-shot premise for single-image depth estimation. Finally, on SUN-RGBD, Depth Anything fairs better in our evaluation setup than in the evaluation reported in the original paper.

**Evaluation metric for sharp boundaries.** For both our depth-based and mask-based boundary metrics, we apply the same weighted-averaging strategy to account for multiple relative depth ratios. F1 values (depth-based metrics) and recall values (mask-based metrics) are averaged across thresholds that range linearly from $t_{min} = 5$ to $t_{max} = 25$. Weights are computed as the normalized range of threshold values between $t_{min}$ and $t_{max}$, such that stronger weights are given towards high threshold values.

# D  APPLICATIONS

Metric, sharp, and fast monocular depth estimation enables a variety of downstream applications. We showcase the utility of Depth Pro in two additional contexts beyond novel view synthesis: conditional image synthesis with ControlNet (Zhang et al., 2023b) and synthetic depth of field (Peng et al., 2022a).

**Depth-conditioned image synthesis.** In this application we stylize an image through a text prompt via ControlNet (Zhang et al., 2023b). To retain the structure of the input image, we predict a depth map from the input image and use it for conditioning the image synthesis through a pretrained depth-to-image ControlNet SD 1.5 model. Figure 12 shows the input image, prompt, and predicted depth maps and synthesis results for Depth Pro, Deoth Anything v2, Marigold, and Metric3D v2. We find that only Depth Pro accurately predicts the cables and sky region, resulting in a stylized image that retains the structure of the input image. Baselines either miss cables, causing the cable car to float in mid-air (Depth Anything v2), or add a gradient to the sky (Marigold).

**Synthetic depth of field.** Synthetic depth of field can be used to highlight the primary subject in a photo by deliberately blurring the surrounding areas. BokehMe (Peng et al., 2022a) introduces a hybrid rendering framework that marries a neural renderer with a classical physically motivated renderer. This framework takes a single image along with a depth map as input. In this context, it is essential for the depth map to delineate objects well, such that the photo's subject is kept correctly in focus while other content is correctly blurred out. Furthermore, the depth map should correctly trace out the details of the subject, to keep these (and only these) details correctly in focus. Figure 13 shows the advantage afforded by Depth Pro in this application. (We keep the most salient object in focus by setting the refocused disparity (disp_focus) hyperparameter of BokehMe as the disparity of the object.)

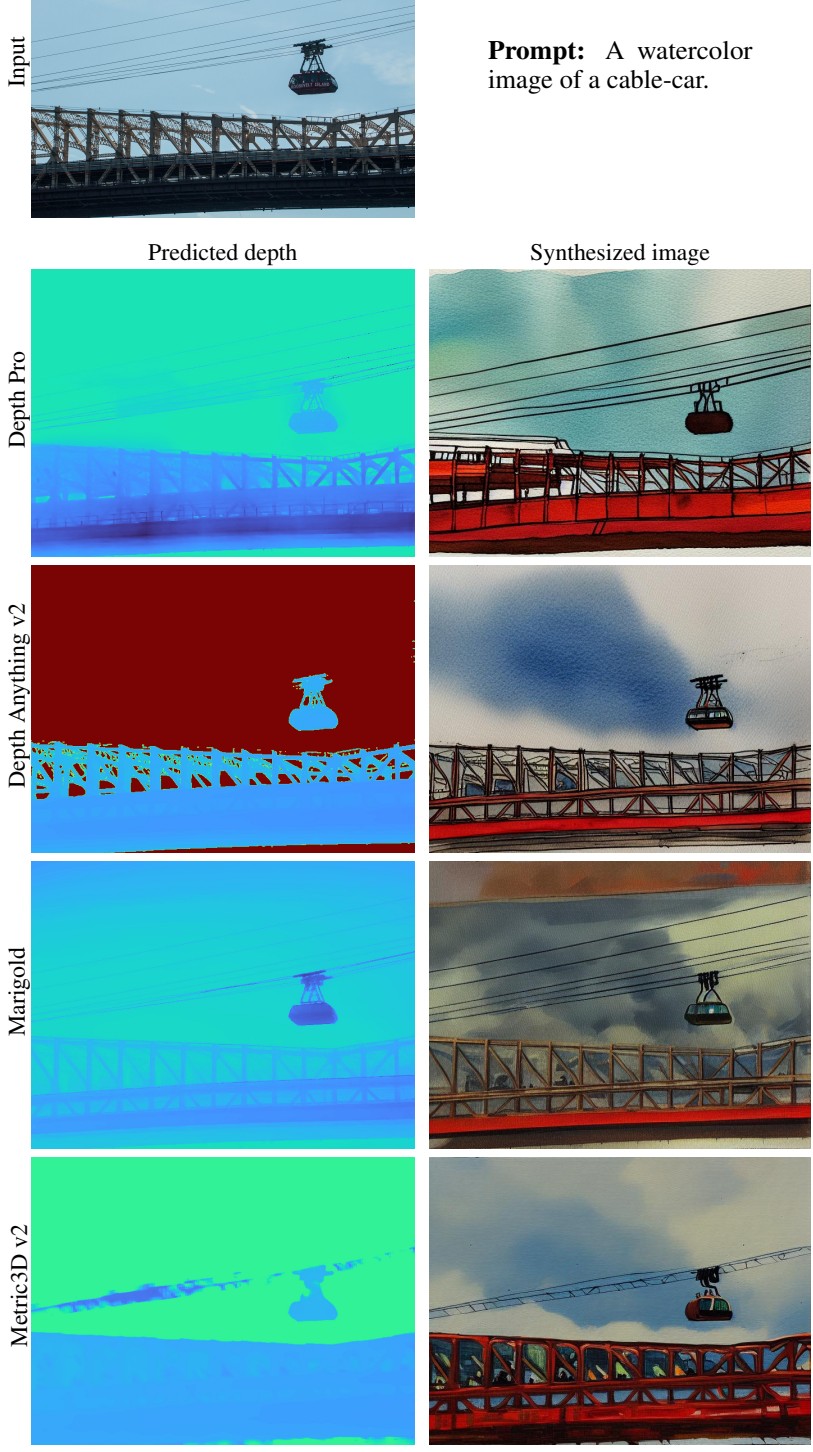

Figure 12: **Comparison on conditional image synthesis.** We use ControlNet (Zhang et al., 2023a) to synthesize a stylized image given a prompt (top row, right) and a depth map. The depth map is predicted from the input image (Li et al., 2022a) (top row, left) via Depth Pro, and baselines. The left column shows depth maps, the right column the synthesized image. For the baselines, missing cables (Depth Anything v2 & Matric3D v2) or a spurious gradient in the sky (Marigold) alter the scene structure of the synthesized image.

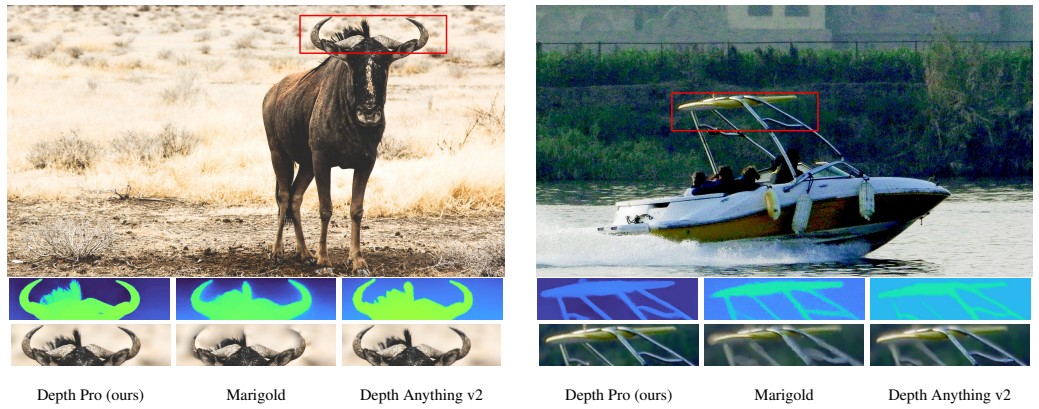

Figure 13: **Comparison on synthetic depth of field.** We compare the synthetic depth of field produced by BokehMe (Peng et al., 2022a) using depth maps from Depth Pro, Marigold (Ke et al., 2024), and Depth Anything v2 (Yang et al., 2024a). Zoom in for detail.

