# OpenReview forum: "Depth Pro: Sharp Monocular Metric Depth in Less Than a Second"
_ICLR.cc/2025/Conference — ICLR 2025 Poster_

### Official Review · Reviewer_85gd · 2024-10-19

**Soundness:** 4
**Presentation:** 4
**Contribution:** 3
**Rating:** 6
**Confidence:** 4

**Summary:**

This paper presents a novel framework for high-resolution metric depth estimation, demonstrating both excellent organization and a well-articulated motivation. The qualitative outcomes are impressive, and the quantitative results also demonstrate the state-of-the-art performance. The authors assert four main contributions: (1) an innovative network architecture, (2) a newly proposed evaluation metric for boundary tracing, (3) newly designed loss functions, and (4) the capacity to estimate camera focal length.

**Strengths:**

- This paper is well-written and motivated.
- The proposed method demonstrates impressive performance.
- It addresses the metric monocular depth estimation problem, a crucial task with significant application prospects.
- The method estimates camera focal length, eliminating the need for image metadata—a practical advancement.
- The experiments detailed in the main text and appendix are comprehensive and solid.

**Weaknesses:**

There are some minor issues:
- There are a few unclear statements regarding Figure 3:
  - Along which dimension does the concatenation operation perform?
  - What is the specific implementation of the Merge operation?
- At Line 234, the statement "produces a feature tensor at resolution 24 × 24 per input patch (features 3 – 6 in Fig. 3)" seems to not coincide with the figure label.

**Questions:**

- Marigold and Depth Anything have demonstrated that using synthetic training data can effectively improve details in depth prediction. Depth Pro also uses synthetic data to enhance high-frequency details in the depth map, yet its predicted boundaries are significantly better than those of existing methods. What is the main reason for this?significantly better than the existing methods. What is the main reason for this?
- Comparative methods such as Depth Anything and Marigold are trained with low-resolution images, while Depth Pro is trained with high-resolution images. Is this the reason why Depth Pro achieves better detail in depth prediction?
- According to Table 13, Depth Pro appears to use more training data than other competitors. Can you compare the scale of training data used by Depth Pro with that of other competitors?
- According to Table 1, Depth Pro has a significant gap with the best performing methods on some datasets (such as ETH3D). Can you analyze the reasons for this discrepancy?

---

> ### Author Response · Authors · 2024-11-27
>
> Thanks for the feedback!
>
> **Figure 3**: The different patches (tiles) in Fig. 3 are concatenated along the batch dimension, so that the resulting input image tensor to the ViT encoder can be batch-processed very efficiently. We clarified this in the updated paper. We also added a more detailed description of the merge operation in Sec. C.1 of the appendices.
>
> **Training data**. Please note that Tab. 13 (now Tab. 15) also contains test datasets. Overall, Depth Pro is trained on approximately 5.1M real and 2.1M synthetic depth maps. This is significantly less than e.g., the 16.2M samples reported by Metric3D v2.
>
> **Performance gap to Marigold**. Our controlled experiments indicate that a high native output resolution is a necessary condition for predicting high-frequency details. This alone is not sufficient however, as demonstrated by Depth Pro outperforming approaches with varying output resolutions. We observe further improvements on boundary metrics through the proposed architecture, loss functions, and training curriculum.
>
> **Varying performance on zero-shot datasets**. The performance of all tested methods varies significantly across evaluation datasets. A possible explanation could be the selection of training datasets, which differs for each method and may result in biases towards certain domains. Hence, for fair comparison we purposefully collect a diverse set of zero-shot test datasets that represents different domains. On average Depth Pro outperforms all other approaches, which indicates a robust performance across diverse domains.

---

> > ### Author Response · Authors · 2024-11-30
> >
> > Dear reviewer 85gd. Please let us know if we have addressed your concerns. If you believe we have, could you please consider raising your score to reflect that?

---

### Official Review · Reviewer_AZEW · 2024-11-02

**Soundness:** 4
**Presentation:** 4
**Contribution:** 4
**Rating:** 8
**Confidence:** 5

**Summary:**

The paper introduces Depth Pro, a model for zero-shot metric monocular depth estimation that generates high-resolution depth maps with precise boundary details at sub-second processing speeds. Key innovations include a multi-scale vision transformer architecture, a mixed training regimen with real and synthetic data for improved accuracy, and a dedicated evaluation approach for boundary sharpness. The model also features state-of-the-art focal length estimation without the need for camera metadata. Depth Pro outperforms prior approaches in accuracy and efficiency, making it suitable for applications like novel view synthesis.

**Strengths:**

- The model demonstrates strong zero-shot performance for metric depth estimation, effectively handling diverse input images without fine-tuning.
- It achieves superior focal length estimation accuracy, which enhances depth map scaling and generalization across different image conditions.
- Comprehensive ablation studies are provided to analyze the impact of key architectural and training choices, showcasing the robustness of the proposed method.
- The experiments are conducted on a wide range of datasets, underscoring the model’s effectiveness and applicability across various scenarios.
- Detailed training procedures are described, allowing for reproducibility and providing insights into the model’s development process.
- The method exhibits higher edge accuracy compared to conventional techniques, maintaining competitive inference times.
- It outperforms state-of-the-art approaches, such as DepthAnything, DepthAnything v2, and Marigold, with clear comparative analysis highlighting these advancements.

**Weaknesses:**

The most critical components that influence performance and boundary accuracy among various technical contributions are not clearly highlighted, leaving ambiguity about which elements contribute most to the model's success.

**Questions:**

1. If the loss function were designed using depth instead of canonical inverse depth, would the results differ? Have you conducted any experiments to compare these approaches?
2. Among the various technical contributions (e.g., curriculum learning, focal length estimation, and the patch encoder with multiple scales), which components most significantly affect the overall performance metrics (e.g., abs_rel, δ_1) and boundary accuracy, respectively? Although multiple technical contributions are mentioned, it is unclear which are the most impactful.
3. Is there any tradeoff between overall performance and boundary accuracy related to certain hyperparameters in the deep neural network or the training loss?
4. For calculating the final depth, does the proposed method use the predicted inverse depth C, the predicted focal length f_px, and the ground-truth width w, as in the equation D = f_px / (wC)?
5. As I understand it, all experiments were conducted with evaluation using metric scale depth, without median-scaling evaluation. Is my understanding correct?
6. Have you conducted experiments on fine-tuning the proposed model (DepthPro)? If so, is there a significant performance improvement between the zero-shot setting and fine-tuning?
7. What is the training time and GPU resources required to train the proposed method?

---

> ### Author Response · Authors · 2024-11-27
>
> Thank you for the feedback!
>
> Please find our response to the question on the most critical components for performance in the global response above and remaining responses below.
>
> **Effect of predicting inverse disparity**. Thanks for the suggestion. To assess the effect of predicting inverse depth, we conducted the following experiment (Tab. 10): We trained a single ViT encoder and DPT decoder to predict inverse depth, depth, and log-depth on Hypersim. We then evaluate the accuracy in predicting depth on several depth ranges. Across all ranges we observe lower errors for predicting inverse depth than for depth or log-depth.
>
> **Tradeoff between boundary accuracy and metric depth**. To shed more light on this tradeoff, we supplement our controlled experiments with boundary evaluation results. For stage 1 losses, our proposed loss configuration yields the best depth accuracy and boundaries. For stage 2 losses, our proposed configuration of first and second order losses balances metric depth and boundary accuracy. Here, removing the second order loss improves metric depth at the cost of less accurate boundaries. Further removing the MSGE loss reverses this effect. For the curriculum we observe improved metric depth by training the two proposed stages. Training only a single stage slightly improves boundaries at the cost of metric depth accuracy.
>
> **Calculating final depth**. That is correct. When the focal length is available however (e.g. through the EXIF data or in the dataset), we will use it instead of the predicted focal length.
>
> **Median-scale depth in experiments**. That is correct. Table 1 and 4 report absolute errors, and we do not use any kind of median scaling or alignment to the ground-truth. Boundary metrics in Table 2 are scale-invariant, which allows us to run comparisons against non-metric methods such as DPT, Marigold or DepthAnything-relative v1 and v2.
>
> **Fine-tuning experiments**. In-domain fine-tuning should almost always improve the in-domain accuracy on downstream tasks, usually at the cost of diminished generalization abilities. As we focus on zero-shot generalization we deemed in-domain fine-tuning experiments beyond the scope of the paper.
>
> **Training time & compute**. Depth Pro trains in approximately 50K GPU-hours on a Volta GPU with FP32 precision.

---

> > ### Comment · Reviewer_AZEW · 2024-12-02
> >
> > Thank you for addressing my questions. The author's response clarified my curiosity.
> >
> > I look forward to seeing this paper presented at ICLR.

---

### Official Review · Reviewer_a8Yt · 2024-11-02

**Soundness:** 3
**Presentation:** 3
**Contribution:** 2
**Rating:** 3
**Confidence:** 4

**Summary:**

The proposed Depth Pro model employs a ViT architecture for zero-shot metric monocular depth estimation, targeting applications such as novel view synthesis. By employing patch-based, multi-scale processing, Depth Pro achieves high-resolution depth predictions while preserving real-time performance and edge sharpness. This paper introduces a two-stage training approach that integrates synthetic and real-world datasets, enhancing depth boundary accuracy. Furthermore, the model includes boundary evaluation metrics and a focal length estimation component, improving its robustness in handling images lacking metadata.

**Strengths:**

1. Structured Training Curriculum: The curriculum is specifically tailored to handle the unique strengths and weaknesses of synthetic and real-world datasets, resulting in a model that generalizes effectively while producing fine boundary details in depth maps.

2. Enhanced Boundary Evaluation Metrics: New metrics for evaluating depth boundaries address a gap in existing benchmarks by focusing on boundary precision, which is critical for applications like view synthesis that demand fine details.

**Weaknesses:**

1. Dependence on pretrained ViT with Limited Architectural Innovation: Depth Pro benefits from pretrained ViT backbones, yet its architecture primarily builds on existing elements rather than introducing fundamentally new mechanisms for depth estimation, which limits its architectural novelty.

2. Heavy Reliance on Synthetic Data for Boundary-Sensitive Training: The model’s second-stage training emphasizes synthetic datasets for boundary sharpness. This reliance on synthetic data could impact generalization to real-world environments, particularly in complex or unstructured settings where boundaries are less distinct.

**Questions:**

See weaknesses.

---

> ### Author Response · Authors · 2024-11-27
>
> Thank you for the feedback!
>
> **Limited architectural innovation**: We propose a novel architecture that leverages pretrained ViT backbones in a highly efficient way - it is more accurate and faster than upscaling existing ViT backbones (see comparison to DINOv2 in Tab. 8, comparison to Metric3D v2 in Tab. 1 & Tab. 5) and is capable of accurately tracing thin structures (Tab. 2). This is in contrast to prior state-of-the-art approaches such as DepthAnything v1 and v2, and Metric v2, which all relied on plain ViT encoders, DINOv2 weights, and a plain DPT decoder.
> Beyond the novel architecture of the depth estimation branch, we propose a simple yet effective approach to handing images without camera metadata. This is in contrast to almost all other approaches to metric depth estimation (which rely on ground truth metadata) and outperforms prior state of the art on focal length estimation (Tab. 3). We further propose a novel metric for assessing the quality of predicted depth boundaries and benchmark prior work. And we demonstrate the importance of accurate depth boundary prediction on three downstream applications.
>
> **Reliance on synthetic data**: Although reliance on synthetic data during training could hypothetically impact generalization, we demonstrate that this is not the case here. Figures 1, 4, 5, 6, 7, 8, 9 show qualitative demonstrations of Depth Pro predicting thin structures in complex environments on unseen real-world images. Quantitative demonstrations come from the zero-shot evaluation of boundary metrics in Tab. 2, which we conduct on several datasets from different domains to specifically assess generalization to unseen environments. The quantitative and qualitative results on matting datasets in particular demonstrate excellent generalization to real-world data outside of the training domains.

---

> > ### Author Response · Authors · 2024-11-30
> >
> > Dear reviewer a8Yt. Please let us know if we have addressed your concerns. If you believe we have, could you please consider raising your score to reflect that?

---

### Official Review · Reviewer_eL31 · 2024-11-03

**Soundness:** 4
**Presentation:** 4
**Contribution:** 4
**Rating:** 10
**Confidence:** 5

**Summary:**

This paper introduces a foundational model for single-image depth estimation, trained on a large collection of datasets. The approach results in high-resolution depth maps with sharp object boundaries and enables the estimation of camera intrinsic parameters. The model outputs metric-scale depth predictions and demonstrates its effectiveness across multiple datasets, outperforming previous methods in both quantitative and qualitative evaluations. The contributions of this work are significant, with the pre-trained model holding substantial potential for a range of downstream tasks.

**Strengths:**

1. High-Resolution Predictions: The model produces depth maps at resolutions as high as 1536 x 1536, attributed to its multi-scale vision transformer architecture.

2. Comprehensive Training: The use of extensive depth estimation datasets for training and testing demonstrates substantial effort in data processing. The pre-trained model is valuable to the community and holds potential for multiple downstream tasks.

3. Camera Intrinsic Estimation: The model can estimate camera intrinsic parameters and shows strong zero-shot generalization due to its training on diverse datasets, which is crucial for real-world scenarios where camera calibration is challenging.

4. Visual and Quantitative Superiority: The qualitative results clearly show better object boundary preservation compared to previous methods, and the quantitative evaluations across datasets highlight the model's excellent zero-shot performance. The evaluation on downstream tasks like image synthesis showcases the model’s adaptability.

**Weaknesses:**

1. Limited Novelty: Although effective, the proposed method's novelty is somewhat constrained as it leverages existing approaches. The loss functions and training strategy draw heavily from DPT (Ranftl et al., 2022), and the metric depth estimation follows ideas from Metric3D (Yin et al., 2023). More discussion on the unique aspects would strengthen the paper.

2. Depth Consistency in Videos: While single-image results are impressive, depth consistency across video frames is not explored. Given the model’s metric-scale depth predictions, it would be valuable to assess its utility in video-based tasks such as visual SLAM, camera pose estimation, and 3D reconstruction.

**Questions:**

See weakness

---

> ### Author Response · Authors · 2024-11-27
>
> Thank you for the feedback!
>
> **Limited novelty**: Please note that our contributions go beyond loss functions and training strategy. As outlined in Sec. 3.1, we propose an efficient architecture that allows high-resolution metric depth prediction with excellent zero-shot capabilities. Our proposed approach reaches higher depth accuracy than alternatives (Tab. 1 & Tab. 9), produces more accurate boundaries (Tab. 2), predicts more accurate focal lengths (Tab. 3) and is more efficient than prior work (Tab. 5). In contrast to almost all other work on single image metric depth prediction, it works on images without metadata by predicting the focal length (Sec. 3.2). We further propose a novel metric for evaluating the quality of predicted depth boundaries (Sec. 3.3).
>
> **Temporal consistency**: We thank eL31 for the suggestion and agree that the exploration of further applications would be interesting. As we focus on single-image depth estimation in this paper and already explore three closely related downstream applications (novel view synthesis, controlled image synthesis, synthetic depth of field) in the Appendix, we politely leave the exploration of multi-frame use cases to future work.

---

### Official Review · Reviewer_yFUY · 2024-11-05

**Soundness:** 3
**Presentation:** 2
**Contribution:** 3
**Rating:** 6
**Confidence:** 2

**Summary:**

The paper introduces a new approaches for estimating a depth map from a single image. The paper claims many advantages of the method over competing approaches, in particular the ability to estimate absolute scale under all conditions (i.e., without metadata), high resolution, and speed. The approach includes a transformer-based architecture, hybrid training approach, and focal length estimation to address scale issues. Extensive experimental results are presented.

**Strengths:**

- Important task (monocular depth estimation)
- impressive visualizations (e.g. figure 1)
- Achieves an averaged best performance on multiple datasets and demonstrated great generalization for open-world images.
- Excels  in distinguish boundaries
- Evaluation across a large number of data sets/ablations. Of particular note is the difficulty of avoid cross-data set contamination (test samples appearing in training, etc.) and restricted access.

**Weaknesses:**

- Method section is not clear enough. Missing model details.
- How are features 3-6 obtained since the images are splits into {1x,3x,5x} 384^2 and none of them matches the feature resolution at 48^2 and 96^2?
- what's the architecture of the patch encoder?
- What's the detailed architecture of the focal length head? How does this module fuse both multi-scale features and the features from the extra "image encoder for local length"? The role of the this estimation piece is critical in understanding how it is possible for the method to achieve absolute depth estimation even when camera parameters are absent but the description is lacking.
- Overall, the writing of this paper is poor. It's hard to follow both the proposed method and the experimental evaluation though the results are impressive.

**Questions:**

- the proposed method excels in detecting boundaries. Do the authors have a sense why this happens? Due to the multi-scale derivative objective? Ablations about the reason are encouraged.
- Fig. 4 is only mentioned in l457-458. No details are provided unless in their caption. It's hard to understand how DepthPro helps novel view synthesis.
- How does the proposed model handle images of an arbitrary scale?
- One claim of this paper is the efficient inference. Running time numbers should be reported. Currently they are only demonstrated in Figure 1.

---

> ### Author Response · Authors · 2024-11-27
>
> **Method not clear / missing details**. We appreciate the feedback and added additional sections on the feature merging (Sec. C.1) and focal length head (Sec. C.2) to the appendices and updated Sec. 3.1. We further updated Figure 3. We summarize here for convenience:
>
> * **Feature merging**: The patch encoder ingests image patches of 384^2 and yields feature patches of 24^2. At the coarsest resolution of 384^2, we directly obtain Feature 5 of size 24^2. The same happens with the image encoder, yielding Feature 6. At the next higher scale, we split the 768^2 image into 3x3 = 9 overlapping patches of 384^2 each and obtain 9 patches of 24^2 each. The patches are merged to Feature 4 of size 48^2. For merging patches we generate a Voronoi partition of the feature map with patch centers as seeds. Each cell then corresponds to the area of a feature patch we can directly copy. Figure 11 in the appendices illustrates the merging. Features 1-3 are merged analogously.
> * **Patch encoder**: Details on the backbone architectures can be found in Sec. B.2 of the Appendix. We studied several backbone variants (see Tab. 7 of the Appendix) and opted for a ViT-L DINOv2 architecture.
> * **Focal length head**: The focal length encoder uses the same architecture and pre-trained weights (ViT-L DINOv2) as the image encoder and the patch encoder. Same as the image encoder, it operates on the full image downsampled to 384^2 and yields a feature tensor of 24^2. To estimate the focal length, we simply add the feature maps from both image encoders and pass them through a 3-layer convolutional network, which predicts the focal length. The network reduces the channel dimension from 128 to 64, to 32, to 1 via kernels of size 3, 3, 6, and strides 2, 2, 1. Convolutions are interleaved with ReLUs. The focal length head does not operate on multi-scale features but only features from the image encoder(s) at a single scale. As this was not clear from Figure 3, we revised it.
>
> **Effect on novel view synthesis**. Thank you for the feedback, we supplemented Fig. 4 with a more detailed explanation. Novel view synthesis requires depth maps to be metric to display objects at the right size and depth boundaries to be well aligned with image edges. Fig. 4 focuses on the artifacts due to misaligned boundaries and missed thin structures. Here, all prior work produce ghosting artifacts as shown in the insets of the horse and zebra images. For the windmill, we further observe thin structures being missed by Marigold and Depth Anything v2.
>
> **Arbitrary image sizes**. We resize images of all sizes to 1536^2 via bicubic interpolation for processing by our network and resize depth predictions to the original input resolution. We found this simple scheme to work robustly across arbitrary image sizes as demonstrated through its strong zero-shot performance for metric depth and boundary accuracy.
>
> **Runtime**. We summarize runtimes in Fig.1, Fig. 2, and report detailed runtime numbers for various input resolutions in Sec. A.3 of the Appendix. To summarize here for convenience: Among all approaches with a fixed output resolution, Depth Pro has the highest native output resolution, processing more than 3 times as many pixels as the next highest, Metric3D v2. Yet Depth Pro has less than half the parameter count and requires only a third of the runtime compared to Metric3D v2. The variable-resolution approaches (PatchFusion and ZeroDepth) have a considerably larger runtime, with the faster model, ZeroDepth, taking almost 4 times as long as Depth Pro, even for small VGA images.

---

> > ### Author Response · Authors · 2024-11-30
> >
> > Dear reviewer yFUY. Please let us know if we have addressed your concerns. If you believe we have, could you please consider raising your score to reflect that?

---

### Author Response · Authors · 2024-11-27

We thank the reviewers for their feedback!

We revised the paper accordingly. As multiple reviewers asked for clarification on the effect of individual contributions on performance, we provide an answer here and respond to other questions after the respective reviews.

**Effect of contributions on performance**. We thank reviewers yFUY and AZEW for their suggestions. The increase in boundary accuracy over prior work can be attributed predominantly to the high output resolution, the loss functions, and the architecture. To better disentangle these factors, we added boundary metrics computed on the validation set of Hypersim to our controlled experiments and conducted an experiment on different output resolutions.

* Tab. 11 demonstrates that our proposed losses for stage 1 (config 1C) double the accuracy on boundaries in comparison to a standard mean absolute error on metric datasets (config 1A). The losses in stage 2 are chosen to balance metric depth accuracy and boundary accuracy (see Tab. 12). The same rationale applies to the training curriculum in Tab. 13 - training with just a single stage (config 3B) results in even sharper boundaries than our proposed configuration 3A but yields less accurate depth predictions on average.
* In Tab. 9 we compare our network architecture to alternatives that also predict at 1536^2. We find that transformer-based architectures fare better than convolutional architectures and among the transformers, ViT-based architectures perform best. Notably, our patch-based processing increases boundary recall on DIS5K by ~20% over scaling up the DINOv2 architecture, and more than 2.8x in comparison to e.g. SWINv2.
* Finally, to assess the effect of the output resolution on UnrealStereo4K by downsampling the ground truth to a specific resolution, upsampling again to the full resolution, and evaluating against the unmodified ground truth. This experiment provides an approximate upper bound on metric depth and boundary accuracy a method could achieve if it predicted the perfect ground truth at a specific resolution. We observe that the effect on boundary accuracy is ~3x per doubling of image dimensions. While this result suggests that predicting at high resolution is a necessary condition for high-quality boundaries, it does not present a sufficient condition. This is demonstrated by the comparison of Depth Pro against variable-size approaches like ZeroDepth and PatchFusion, which predict at a higher output resolution but fare worse in boundary metrics.

---

### Meta-Review · Area_Chair_UTb8 · 2024-12-21

**Metareview:**

This paper received mostly positive ratings from reviewers. The concerns raised by reviewers were effectively addressed in the rebuttal, leading to general satisfaction with the authors’ responses. Reviewer yFUY raised issues regarding poor writing, unclear technical details, and insufficient runtime comparison. The authors provided more detailed explanations of technical aspects and clarified runtime comparisons in their response. While these improvements addressed the reviewer’s concerns, the area chair encourages the authors to further enhance the clarity and quality of the writing in the final version. Reviewers eL31 and a8Yt expressed concerns about the limited novelty of the work. The authors clarified the contributions of their work in the rebuttal, particularly emphasizing the innovative aspects of their approach. Reviewer AZEW noted that the critical components influencing performance and boundary accuracy among the technical contributions were not clearly highlighted. The authors included additional ablation studies in their rebuttal, which helped clarify the importance of specific components. The reviewer acknowledged that their concerns were addressed. Reviewer 85gd requested a better explanation for the strong performance of the proposed method. The authors provided a more comprehensive explanation of the factors contributing to the method’s performance, which satisfied the reviewer. Given the authors’ effective rebuttal and the reviewers’ satisfaction with the addressed concerns, this paper makes a valuable contribution to the field and is recommended for acceptance.

**Additional Comments On Reviewer Discussion:**

Reviewer a8Yt expressed concerns regarding limited architectural innovation and the generalization to real-world environments. The rebuttal effectively addressed these issues by explaining the novel architecture of the depth estimation branch, emphasizing the contribution of a simple and effective method of handling images without camera metadata, and demonstrating excellent generalization to real-world data outside the training domains. Reviewer a8Yt did not provide further comments, and the area chair finds the rebuttal convincing in addressing the raised concerns.

---

### Decision · Program_Chairs · 2025-01-22

Accept (Poster)